# The Narrow Gate: Localized Image-Text Communication in Native Multimodal Models

**Alessandro Pietro Serra**[1,2*]  **Francesco Ortu**[1,3*]  **Emanuele Panizon**[1*]

**Lucrezia Valeriani**[1,3]  **Lorenzo Basile**[1,3]  **Alessio Ansuini**[1]

**Diego Doimo**[1†]  **Alberto Cazzaniga**[1†]

[1] Area Science Park, Trieste, Italy  [2] SISSA, Trieste, Italy  [3] University of Trieste, Trieste, Italy

{diego.doimo, alberto.cazzaniga}@areasciencepark.it

## Abstract

Recent advances in multimodal training have significantly improved the integration of image understanding and generation within a unified model. This study investigates how vision-language models (VLMs) handle image-understanding tasks, focusing on how visual information is processed and transferred to the textual domain. We compare *native multimodal VLMs*, models trained from scratch on multimodal data to generate both text and images, and *non-native multimodal VLMs*, models adapted from pre-trained large language models or capable of generating only text, highlighting key differences in information flow. We find that in native multimodal VLMs, image and text embeddings are more separated within the residual stream. Moreover, VLMs differ in how visual information reaches text: non-native multimodal VLMs exhibit a distributed communication pattern, where information is exchanged through multiple image tokens, whereas models trained natively for joint image and text generation tend to rely on a single post-image token that acts as a *narrow gate* for visual information. We show that ablating this single token significantly deteriorates image-understanding performance, whereas targeted, token-level interventions reliably steer image semantics and downstream text with fine-grained control.

## 1 Introduction

The rise of foundation models [1] trained on vast amounts of text has transformed natural language processing (NLP), showing that a single large language model (LLM) [2] can handle many different linguistic tasks [3–5]. The rich set of features encoded in LLM embeddings has been then used as an effective prior knowledge both for text-conditional image generation [6–8] and image understanding [9–13]. Recently, the availability of large open datasets [14, 15] and improved techniques to align text and image embeddings [16] have also enabled the creation of multimodal models that can understand and generate visual content within a single architecture [17–20]. This unification allows a deeper understanding of the visual world, as generative tasks often require insight into the fundamental concepts and relationships within the data [21]. For example, a model that generates images from text descriptions must grasp the semantic content of those images to ensure that they faithfully reflect the details and intent of the text [22, 23]. As a result, research has rapidly progressed in

---

[*]Equal contribution

[†]Equal supervision.

39th Conference on Neural Information Processing Systems (NeurIPS 2025).

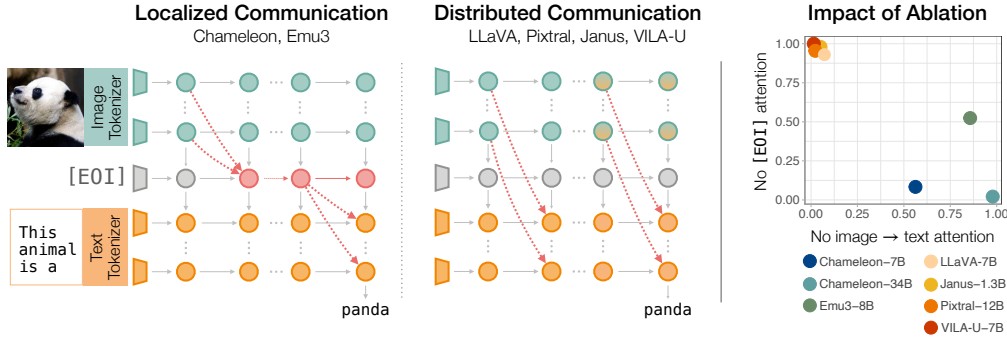

Figure 1: **Image-Text Communication in Vision Language Models.** The figure compares how different VLM architectures handle image-text information flow. **Left**: Native multimodal models (Chameleon, Emu3) process visual and textual tokens separately with information transfer occurring primarily through a single *narrow gate* - the end-of-image token (`[EOI]`). **Center**: In non-native multimodal models (LLaVA, Pixtral, Janus, VILA-U), communication is distributed across many internal image tokens; visual tokens in deeper layers align more strongly with text tokens. **Right**: The figure shows the relative performance on MS-COCO image captioning after performing different ablations, compared to no ablation (see section 3.4). The results show that for narrow gate models, removing attention to the `[EOI]` token (y-axis) impacts performance more significantly than removing attention to all image tokens (x-axis).

integrating multiple modalities into a unified framework with an increasingly deeper multimodal fusion. Early approaches used cross-attention modules between modality-specific encoders [17]. Further advancements highlighted the importance of using a pre-trained LLM backbone combined with lightweight projection layers, often fine-tuning the LLM to refine multimodal representations [19, 24, 25]. More recently, research has shown success in training vision language models (VLM) from scratch [20, 26], achieving performance similar to text-only output VLMs on visual understanding tasks. In this work, we use the term *native multimodal* to refer to VLMs that are trained from scratch on both text and image modalities, without relying on a pretrained LLM backbone and CLIP-based text-aligned vision encoder, and capable of generating both images and text.

Although recent studies have examined the internal mechanisms of unimodal output VLMs [27–30], it remains largely unknown how modalities interact in the hidden representations of native multimodal models. In this work, we focus on some of the latest native VLMs, Chameleon [20] and Emu3 [26], and compare them with a range of *non-native* ones, including both text-only VLMs such as LLaVA [11] and multimodal generators that rely on a pretrained language backbone like VILA-U [31], on how they transfer information from the visual domain to the textual domain in various image understanding tasks.

We observe that in native multimodal models (Chameleon and Emu3), image and text representations remain well-separated in different clusters from input to output, while for non-native models they tend to mix in late layers (section 3.1). Additionally, in this second class of models, visual information flows to text through multiple tokens. In contrast, native multimodal VLMs channel the global image information into a single token placed immediately after the image, the *end-of-image token*, denoted by `[EOI]` (section 3.2). The `[EOI]` token behaves like a memory token [32] or *narrow gate* through which the image information must pass to guide text generation.

We establish the functional role of `[EOI]` by showing that (**1**) blocking attention from text tokens to `[EOI]` causes a sharp drop in performance across classification, VQA, and captioning tasks (section 3.4) and that (**2**) editing the representation of `[EOI]` alters the semantic content of the generated text (section 4.1), confirming its causal role in mediating image semantics. These properties do not appear in non-native multimodal VLMs, where the information between modalities flows through many distributed image tokens. Finally, we propose a fine-tuning strategy to remove the narrow gate in native multimodal models by redistributing visual-to-text transfer across many image tokens (section 4.2).

## 2 Background and Methods

**Model architectures.** We analyze six VLMs differing in their training objectives and multimodal integration. Two of them, Chameleon [20] and Emu3 [26], are native multimodal models trained from scratch to jointly generate images and text using a unified discrete tokenizer with a VQ-GAN image encoder. Since the released Emu3 variants were specialized either for image generation (Emu3-Gen) or for understanding (Emu3-Chat), we fine-tune the Emu3-Gen checkpoint on a balanced mixture of image–text datasets so that it can handle both tasks within a single model (see appendix B.3). This enables a direct comparison with Chameleon, which natively supports the joint generation of images and text. As non-native multimodal VLMs, we analyze two text-output VLMs: LLaVA-onevision-7B [33] (LLaVA) and Pixtral-12B [13] (Pixtral), fine-tuned on Qwen2-7B and Mistral Nemo 12B, respectively. Additionally, we also analyze Janus-1.3B [34] (Janus) and VILA-U [31], which are both multimodal in output and fine-tuned on a pre-trained LLM, DeepSeek-LLM-1.3B [35] and LLaMA-2-7B, respectively [36]. Janus employs a SigLIP vision encoder and separate pathways for image understanding and generation, while VILA-U combines CLIP-based visual features with a VQ-VAE discretization stage, yielding semantically rich visual embeddings. All the models are decoder-only and trained with next-token prediction loss. In Chameleon, Emu3, Janus, and VILA-U the loss is computed on both image and text tokens, while in LLaVA and Pixtral, it applies only to text tokens. In all the cases, a special end-of-image token ([EOI]) marks the transition from image to text. Additional architectural details are provided in table A2, and related multimodal training strategies are discussed in appendix A.

**Datasets.** We study visual understanding tasks where the model is asked to answer questions from VQAv2 [37], generate image captions for Flickr30k [38] and MS-COCO [39], and complete simple prompts about ImageNet images [40]. We provide the experimental setup, including dataset composition, prompt usage, and data selection criteria, in the appendix B.

### 2.1 Analytical Tools

**Analyzing information flow in VLMs.** To understand how VLMs process information, we analyze both attention maps and token representations across layers. We denote the *residual stream* representation of the token at position $i$ at layer $l$ as $x_i^l$. In our setup, images are typically followed by textual instructions. To study how these modalities interact, we focus on the cross-modal attention: let $N_{[\text{EOI}]}$ be the position of the end-of-image token, and $A_{i,j}$ the attention from token i to token j. We define $A_{\text{text}\to\text{img}} = A_{i,j}$ with $N_{[\text{EOI}]} < i \leq N, 0 < j < N_{[\text{EOI}]}$, which captures how much text tokens attend to visual tokens in the sequence.

**Quantifying cross-modal attention.** We construct a metric that quantifies the average attention that all the text tokens give to a token at position $j$ within the image part of the prompt, which we assume to span the first $N_{[\text{EOI}]}$ tokens. Formally, we define the (relative) *cross-modal attention* $f_j^l$ as

$$f_j^l = \frac{1}{C} \frac{1}{|\mathcal{H}|} \sum_{h \in \mathcal{H}} \sum_{i > N_{[\text{EOI}]}} A_{i,j}^{l,h} \tag{1}$$

Where $j = 0, 1, \ldots, N_{[\text{EOI}]}$, $l$, $h$ identify, respectively, the layer and head. $C$ is a normalization factor such that $\sum_{j \leq N_{[\text{EOI}]}} f_j^l = 1$.

**Blocking Cross-Modal communication with Attention Knockout.** We use the *attention knockout* [41] to selectively block attention between tokens. Given a set of source tokens $\mathcal{S}$ and a set of target tokens $\mathcal{T}$, we zero out the attention weights $A_{i,j}$, where $i \in \mathcal{S}$ and $j \in \mathcal{T}$ across *all heads* in selected layers to prevent the target tokens from attending to the source tokens.

**Probing the semantic information with Neighborhood Overlap.** To evaluate how well the residual stream encodes an abstract property of the image, like the identity of the main object represented (ImageNet) or the important parts of the scene (MS-COCO), we use the *neighborhood overlap* [42]. The neighborhood overlap measures the consistency between the $k$-nearest neighbors of a datapoint $i$ in the residual stream at a layer $l$, denoted by $\mathcal{N}_k^l(i)$, and the nearest neighbors in a reference data representation, $\mathcal{N}_k^{gt}(i)$, which encodes the abstract property of interest. In the case of ImageNet, all the points belonging to the same class are considered nearest neighbors [42] for the

purpose of computing the reference $\mathcal{N}_k^{gt}(i)$. The *neighborhood overlap*, denoted by $\chi_k^{l,gt}$, can be then written as:

$$\chi_k^{l,gt} = \frac{1}{nk} \sum_{i=1}^n \left| \mathcal{N}_k^l(i) \cap \mathcal{N}_k^{gt}(i) \right| \tag{2}$$

where $|\cdot|$ denotes the cardinality of a set, and $n$ the dataset size. In our experiments, we set $k = 30$.

**Modifying the semantic information with localized interventions.** To evaluate the influence of specific components on model outputs, we use *activation patching* [43, 44]. This approach involves two forward passes on two different inputs: We first collect the activations $\hat{x}_i^l$ from a target input, then replace the corresponding activations $x_i^l$ in a base input during a second forward pass. The impact is assessed by comparing the resulting output distribution $q_{\text{base}}^{\text{patched}}$ to the target output distribution $p_{\text{target}}$. We use a variant of the Jaccard index [45] to quantify the similarity between $q_{\text{base}}^{\text{patched}}$ and $p_{\text{target}}$:

$$\text{Similarity}(q_{\text{base}}^{\text{patched}}, p_{\text{target}}) = \sum_i \min(q_i, p_i) \tag{3}$$

The score ranges from 0 (no overlap) to 1 (identical distributions), measuring how much the patched model mimics the target behavior.

**Reproducibility.** We used the HuggingFace implementations of Chameleon [46, 47], Emu3-Gen [48], LLaVA [49], Pixtral [50], VILA-U [31] and Janus [51] models. We run all the experiments on a single NVIDIA A100 GPUs with 40GB VRAM. For the Chameleon 34B, we used two 40GB GPUs. The code needed to reproduce the experiments is available at: ritareasciencepark.github.io/Narrow-gate.

## 3 Results

### 3.1 Modality Gap in Native Multimodal VLMs

Native multimodal VLMs generate both image and text tokens using a shared transformer backbone, enabling unified next-token prediction across modalities within a common representation space. In contrast, text-output VLMs produce only textual tokens, using image tokens solely as contextual input. Although both VLM types encode image and text tokens in the residual stream, their role in the generation processes differ significantly. This motivates our first analysis of whether native multimodal models develop modality-specific subspaces in the residual stream that separate image and text representations.

**Cross-modal angular separation in VLMs.** To address this question, we randomly select $10,000$ image-caption pairs from the Flickr30k dataset and extract the residual stream representations of all hidden layers at a randomly chosen *text token position* (see appendix B for the complete prompt structure). To compare the geometric organization of the different modalities, we first measure the median cosine similarity between representations of image and text at each hidden layer.

As shown in figure 2-left, this analysis reveals different trends across architectures. In the Chameleon-7B (blue) and Chameleon-34B (green), the representative vectors of image and text tokens remain nearly orthogonal throughout the hidden layers, with median cosine similarity values consistently below $0.10$. The cosine similarity in LLaVA (yellow) and Emu3 (green) is higher, starting at $0.2$ in the initial layers of the network. But while in Emu3, the cosine similarity decreases to $0$ in the last layers of the network, in LLaVA it rises to $0.5$. This implies that in Emu3, the two modalities remain separated in late layers, and in LLava, they increasingly mix. Similarly, other non-native multimodal models (Janus, Pixtral, VILA-U) show the same qualitative behavior of LLaVA and are reported to appendix C.1.

**Clustering of visual and textual embeddings.** To characterize the degree of mixture of the modalities from a different perspective, taking into account the norm of the embeddings, we cluster the embeddings with the Advanced Density Peaks algorithm [52] (see appendix B.4 for a brief introduction) and analyze the composition of each cluster. Figure 2-right shows the homogeneity score [53] of the clusters, measuring how the clustering of tokens aligns with their modality. In

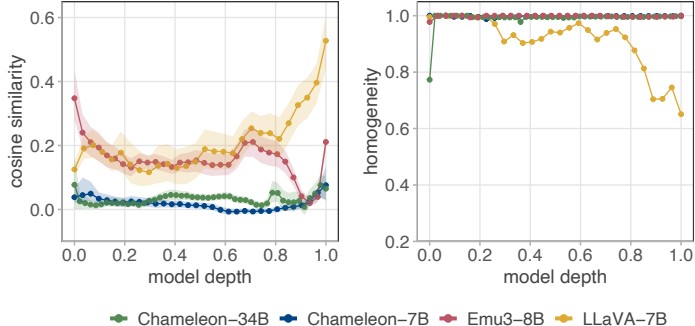

Figure 2: **Modality Gap in VLMs.** (**left**) Cosine similarity between text and image token embeddings shows that LLaVA achieves increasing alignment with depth, while Emu3 shows little alignment and Chameleon maintains orthogonality. Points represent median cosine similarity, with shaded areas indicating the inter-quartile range. (**right**) Homogeneity score assesses how well token clusters (via Advanced Density Peaks) correspond to their original modality.

Chameleon and Emu3 models, the homogeneity score is always $1$, meaning that each cluster contains embeddings from a single modality. In contrast, LLaVA's homogeneity score decreases from almost $1$ to approximately $0.6$ in the later layers of the network, indicating a tendency for the text and image embeddings to mix. In the appendix C.1, we show that, similarly to LLaVA, also in Janus, Pixtral, and VILA-U, the clusters of late layers contain embeddings from both modalities.

We conclude that textual and visual representation spaces are largely separated in the Chameleon and Emu3 models, which are the only ones trained from scratch on multimodal data and can generate both images and textual samples. This brings forth the problem of finding how and where communication between the two modalities is performed.

### 3.2   Analysis of Cross-Modal Attention

As shown in section 3.1, hidden representations of visual and textual tokens mix in the late layers of non-native multimodal models, whereas they occupy well-separated regions of the residual stream in native multimodal models. Despite this structural difference, all model families achieve strong performance on image-understanding tasks, suggesting that cross-modal attention effectively transfers semantic information from visual to textual tokens. In this process, attention matrices play a central role: while enabling information exchange across modalities, they must bridge the modality gap, which is particularly pronounced in native multimodal models such as Chameleon and Emu3. These observations motivate the definition of two key properties for tokens that mediate cross-modal communication: *(i)* having a high weight in text-to-image attention and *(ii)* encoding rich semantic information about the visual input. In the following, we examine whether Chameleon-7B, Emu3, and LLaVA (the latter representing non-native multimodal models) contain tokens that satisfy **both** criteria, identifying candidates that act as communication *gates* between modalities. Corresponding analyses for other models are provided in figure A3.

**Cross-modal attention patterns.**   To quantify the role of each token in image-to-text semantic communication, we use the ImageNet dataset and select 100 images per class for 100 animal classes [40]. We construct $10,000$ prompts of images followed by text of the form "⟨image⟩ `This animal is a __`". The sentence was chosen to guide the model towards generating a class-relevant response.

Figure 3 illustrates the relative text-on-image attention, as defined in section 2.1, at different token positions for Chameleon-7B (left), Emu3 (center), and LLaVA (right). We single out tokens with an average value larger than $1\%$. The remaining tokens are aggregated as *internal image*. Both in Chameleon and Emu, the special `[EOI]` token captures a large amount of cross-modal attention (shown in green in the figure). In Chameleon, this token alone receives $40\%$ to $50\%$ of the total attention from the textual tokens between layer 2 and 6, remaining above $15$ to $20\%$ in the second half of the network. Similarly, in Emu3 `[EOI]` receives more than $30$ to $40\%$ of text-on-image attention after layer 10. Other highly attended tokens are, in Chameleon, the 32nd image token from layer 5 to layer 10 (orange) and the last token of the image receiving between $66\%$ and $86\%$ in the middle and

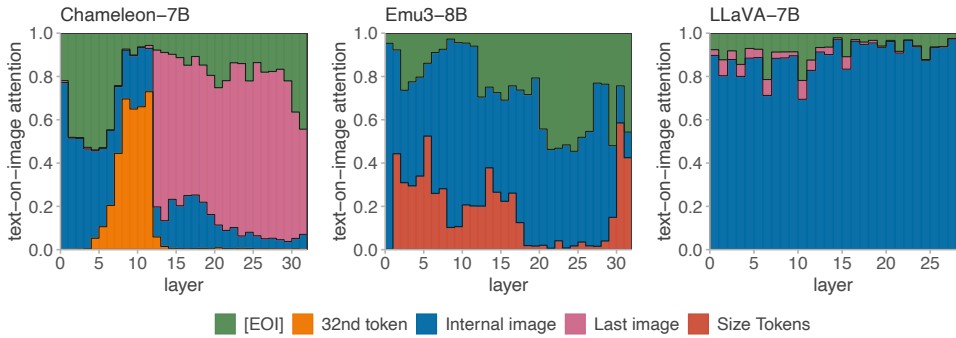

Figure 3: **Cross-Modal Attention Contributions of Image Tokens.** Contribution of different image token positions to the total text-on-image attention across layers in Chameleon (**left**), Emu (**center**), and LLaVA (**right**), computed on ImageNet data. Tokens with an average contribution larger than $1\%$ are singled out. The remaining tokens are aggregated as "internal image".

final layers (shown in pink). In Emu instead, a small group of tokens describing the size of the image receive from 10 to 40% of the total cross-modal attention before layer 17 and after layer 30 (shown in dark orange). The remaining attention is distributed among the other $1024$ image tokens. In the right panel, we present the distribution of average attention among tokens in LLaVA. The [EOI] token accounts for approximately $10 - 20\%$ of the attention between layers 0 and 13, and less than $10\%$ for the remaining layers. All the remaining attention is distributed across the other image tokens. These results suggest the emergence of two distinct communication regimes. Native multimodal models tend to concentrate text-on-image attention on a few single tokens (such [EOI]), which may act as a privileged channel for visual information. In contrast, non-native multimodal models allocate little attention to [EOI], with attention instead dominated by internal image tokens, suggesting a more diffuse and distributed communication between modalities.

## 3.3 Probing the Semantic Content of Visual Tokens

To determine whether the tokens that receive the highest text-on-image attention also encode abstract information about the visual input – the second condition for serving as communication channels – we probe the semantic content of the hidden representations. For this purpose, we use the neighborhood overlap ($\chi_k^{l,gt}$, see section 2.1) with two different reference ground truths: the labels of the ImageNet dataset and the caption embeddings of a Qwen2-7B-GTE text encoder for the MS-COCO.

**Alignment to ImageNet class labels.** First, we measure the neighborhood overlap $\chi_k^{l,gt}$ between image-token embeddings and ImageNet class labels (section 2.1), averaging across positions when multiple tokens are analyzed. As shown in figure 4, the [EOI] token in Chameleon-7B and Emu3 exhibits a sharp rise in $\chi_k^{l,gt}$ after the first few layers, exceeding $0.4$ and $0.25$ respectively, and retaining high values through the model's depth. In contrast, the 32nd, final, and size tokens maintain values near zero, indicating that they attract attention but do not encode meaningful semantic content. The internal image tokens (blue curves) initially capture visual semantics in early layers but gradually lose this information beyond layer 10, leaving [EOI] as the dominant semantic carrier. This pattern holds across scales: in Chameleon-34B, [EOI] exceeds $\chi_k^{l,gt} > 0.4$ after layer 16, while other tokens remain below 0.25 (figure A3). For LLaVA, $\chi_k^{l,gt}$ for [EOI] starts around 0.2 and decreases below 0.1 in later layers, whereas internal image tokens maintain high and stable overlap values ($> 0.4$). A similar trend is observed in other non-native multimodal models, as is shown in figure A3 models such as Janus, Pixtral, and VILA-U: internal image tokens retain the highest $\chi_k^{l,gt}$, while [EOI] contributes little semantic signal. These results suggest two distinct encoding regimes. In native multimodal models (Chameleon, Emu3), the [EOI] token both attracts the majority of text-on-image attention and carries the richest visual semantics, fulfilling the conditions for a *narrow communication gate*. In non-native multimodal models, semantic content remains broadly distributed across internal image tokens, indicating a diffuse, multi-token communication pattern.

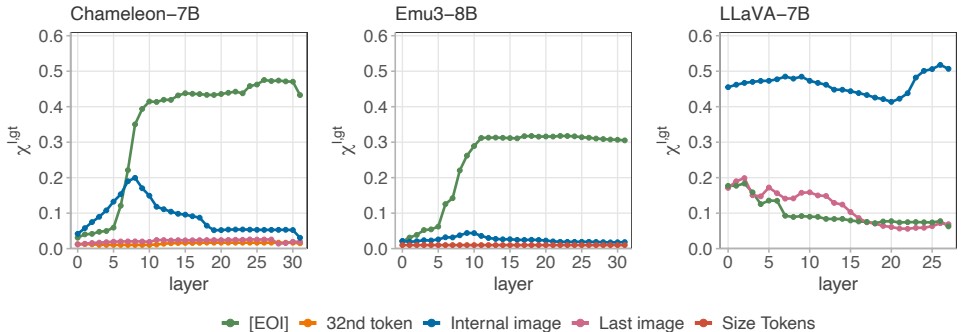

Figure 4: **Localization of Visual Semantic Information.** The figure compares how visual semantic information is localized in Chameleon, Emu3, and LLaVA, by measuring the neighborhood overlap between image tokens and ImageNet labels. The blue curves show the average overlap across all image tokens, excluding the 32nd token for the case of Chameleon.

**Alignment to the embedding image captions.** To verify that this behavior is not specific to classification, we repeated the analysis on MS-COCO captioning data. We compute the neighborhood overlap between the visual-token representations and their corresponding captions. To define a metric in the "space of captions", we use their semantic embeddings through a Qwen2-7B-GTE text encoder (see appendix C.2). The results (figures A2 and A3) mirror those on ImageNet: in native multimodal models, [EOI] maintains the strongest alignment with caption embeddings, whereas in non-native multimodal models, semantic information remains distributed across internal image tokens.

## 3.4 Ablation Experiments: The Effect of Localized Communication on Downstream Tasks

Building on the previous analyses, where we identified the tokens most involved in visual–textual communication, we now test their functional importance through attention ablations. Specifically, we evaluate how blocking communication at selected token positions affects performance on three standard vision–language benchmarks: VQAv2 [37], Flickr30k [38], and MS-COCO [39]. Details on evaluation metrics are provided in appendix B.

**Ablation setup.** For each model, we randomly sample 2,000 examples per dataset and measure baseline performance before applying targeted attention knockouts. The performance for visual question answering VQAv2 is evaluated with the VQA metric [54], for the captioning benchamarks instead we used CIDEr [55]. We block (*i*) communication between the [EOI] and text tokens by zeroing out $A_{\text{text}\rightarrow\text{[EOI]}}$, and (*ii*) all text-on-image attention by zeroing out $A_{\text{text}\rightarrow\text{img}}$ (section 2.1). This isolates the impact of the [EOI] token relative to the broader visual stream. Results for Chameleon, Emu3, and LLaVA are summarized in table 1; additional models (Pixtral, Janus, VILA-U) and further token positions are reported in appendix C.4.

Table 1: **Effect of Attention Knockout on Image Understanding Tasks**. Performance of the models on VQAv2, Flickr-30k, MS-COCO, and ImageNet under different ablation settings. Numbers in bold mark the worst performance for each model and task.

| Model | Ablation | VQAv2 | MS-COCO | Flickr | ImageNet |
|---|---|---|---|---|---|
| | - | 0.51 | 0.48 | 0.34 | 0.46 |
| Chameleon-7B | text → [EOI] | **0.25** | **0.04** | **0.04** | **0.01** |
| | text → img | 0.40 | 0.27 | 0.17 | 0.47 |
| | - | 0.57 | 0.63 | 0.29 | 0.35 |
| Emu3 | text → [EOI] | 0.48 | **0.33** | **0.13** | **0.24** |
| | text → img | **0.42** | 0.54 | 0.21 | 0.30 |
| | - | 0.80 | 0.98 | 0.70 | 0.5 |
| LLaVA | text → [EOI] | 0.80 | 0.97 | 0.71 | 0.45 |
| | text → img | **0.00** | **0.01** | **0.02** | **0.05** |

**Results.** In native multimodal models, ablating the `[EOI]` token causes a substantial performance collapse across all tasks. For instance, in `Chameleon-7B`, accuracy drops from $0.51$ to $0.25$ on VQAv2 and from $0.34$ to $0.04$ on Flickr30k, with similar reductions exceeding $90\%$ on MS-COCO and near-zero overlap on ImageNet. `Emu3-8B` shows comparable degradation, with performance falling by roughly $50\%$ across tasks. In both models, blocking all text-on-image attention ($A_{\text{text}\rightarrow\text{img}}$) produces smaller declines than ablating the single `[EOI]` token, confirming its dominant role. Similar trends hold for Chameleon-34B (appendix C.4). In contrast, non-native multimodal models rely on distributed communication. In LLaVA, performance is unaffected by the ablation of `[EOI]` but collapses entirely when all text-on-image attention is removed ($0.00$ on VQAv2, $0.02$ on Flickr30k, $0.01$ on MS-COCO, $0.05$ on ImageNet). Pixtral, Janus, and VILA-U exhibit the same pattern, indicating that their cross-modal information flow is mediated by many internal image tokens rather than a single position. These ablation results reinforce the distinction between communication regimes. Native multimodal models (Chameleon, Emu3) depend on a localized, token-level bottleneck—the `[EOI]` token—for transferring visual semantics to text. By contrast, non-native multimodal models (LLaVA, Pixtral, Janus, VILA-U) exhibit distributed communication: information flows through numerous image tokens and cannot be disrupted by a localized ablation.

## 4 Implication of Narrow Gate for Model Editing and Robustness

### 4.1 Steering Image Semantics Through Activation Patching

The previous analyses showed that in native multimodal models (Chameleon, Emu3), the `[EOI]` token encodes global visual semantics (section 3.2) and acts as the main communication channel to text (section 3.4). We now test whether this localized structure also enables *constructive* interventions, specifically, whether modifying the representation at `[EOI]` can steer the model's interpretation of an image toward a desired semantic class.

**Activation patching setup.** To do so, we select 20 ImageNet animal classes and sample 100 images per class, forming 10 class pairs. For each pair, we perform activation patching (section 2.1) with the following procedure: (*i*) extract the `[EOI]` representation at each layer $l$ from the *target class* images ($[\text{EOI}]^{l,\text{target}}$); (*ii*) inject this representation into the residual stream of the *base class* images by replacing $[\text{EOI}]^{l^*,\text{base}}$ with $[\text{EOI}]^{l^*,\text{target}}$ at a chosen layer $l^*$; and (*iii*) evaluate the model's output distribution for the final text token in the prompt "⟨image⟩ This animal is a __". We measure the similarity between output probabilities (see equation (3)) and the fraction of cases where the target class probability exceeds that of the base class. For further details on experimental setup see appendix B.

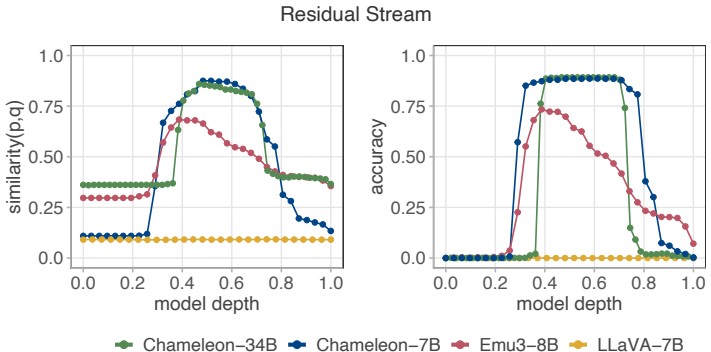

Figure 5: **Impact of Activation Patching at** `[EOI]`**.** Results of patching the `[EOI]` representation from a target class onto the `[EOI]` representation on a base class. The experiment is performed at each layer. Two metrics are evaluated: (**left**) The similarity measure (defined in equation (3)) quantifies how the probability distribution over the vocabulary of the patched image aligns with that of the target. (**right**) The accuracy, defined as the fraction of patched images where the probability of the target class token is larger than the probability of the base class token.

**Results.** Figure 5-left reports the similarity measure as a function of the patched layer for Chameleon-7B, Chameleon-34B, Emu3, and LLaVA. In native multimodal models, patching [EOI] representations from the target class induces strong semantic steering: the similarity rises sharply after a relative depth of $\sim 0.3$, peaking at $0.86$ in Chameleon-7B/34B and $0.70$ in Emu3. The right panel quantifies the success rate of semantic transfer: patching changes the predicted class from base to target in approximately $90\%$ of cases for Chameleon and $75\%$ for Emu3. In contrast, LLaVA, representative of non-native multimodal models, shows no measurable effect. [EOI] patching leaves both similarity and class probabilities unchanged: the base class consistently dominates the output, indicating that cross-modal information is distributed across multiple image tokens rather than localized in a single position. These results demonstrate that in native multimodal models, the [EOI] token provides a causal handle for controlling image semantics: altering its representation is sufficient to steer model predictions. In non-native multimodal models, by contrast, semantic information is distributed throughout the image representation, rendering localized interventions ineffective.

## 4.2 Improving Model Robustness with Localized Fine-tuning

The presence of a narrow gate in native multimodal models can constrain how the model integrates visual and linguistic features, making its representations overly dependent on a specific internal pathway. To promote a richer and more distributed exchange between modalities, in this section, we show a fine-tuning strategy that masks the [EOI] token during training, forcing the model to encode and communicate visual information stored in [EOI] through alternative internal tokens. Details about the training setup are reported in appendix B.3.

**Masked fine-tuning.** Figure 6 shows how model performance evolves during training on 2000 test samples from the MS-COCO and VQAv2 tasks. Solid lines report fine-tuning runs with [EOI] masked, while dashed lines correspond to standard fine-tuning where all input tokens are attended. Purple profiles show the performance when [EOI] is ablated, while green profiles show the test performance when all tokens are visible. At the beginning of training, ablating [EOI] causes a large performance drop, especially for the COCO captioning task (see section 3.4). However, when [EOI] is masked during fine-tuning, performance on both MS-COCO and VQAv2 (solid-purple profiles) increases and approaches the "unablated" performance (green profiles), which remains approximately constant during training. In contrast, when [EOI] is not masked during training (dashed purple profiles), the performance gap and the narrow gate phenomenon remains, since the model continues to rely mostly on [EOI] to store and communicate visual information.

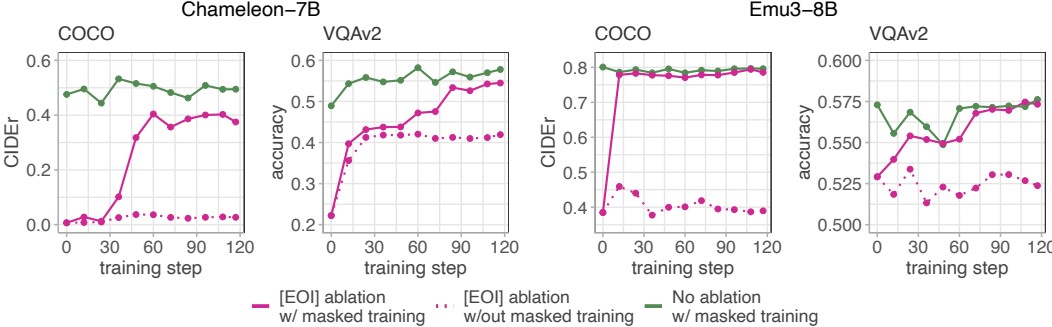

Figure 6: **Performance Dynamics During Masked [EOI] Fine-tuning.** Masking the [EOI] token during fine-tuning reduces the model's reliance on a single narrow gate and encourages a more distributed communication between image and text. Solid lines indicate runs where [EOI] is masked; dashed lines correspond to standard fine-tuning. Green curves report test accuracy with all tokens visible, while purple curves show performance when [EOI] is ablated. When trained with masked [EOI], the performance under ablation (solid purple) progressively recovers and approaches the unablated baseline (green), indicating improved robustness and reduced dependence on [EOI].

# 5 Discussion and Conclusion

**The Role of** `[EOI]` **in Cross-Modal Communication.** In this work, we analyzed information flow within transformer-based VLMs using geometric and mechanistic interpretability techniques. Our study focused on native multimodal models - those trained from scratch to generate both images and text - filling a gap in the existing literature, mainly focused on text-output or fine-tuned multimodal systems [56].

Our analyses reveal a *narrow gate* communication mechanism in native multimodal VLMs. These models exhibit a pronounced modality gap: visual and textual representations remain well separated throughout the network (section 3.1), yet semantic transfer occurs effectively through a *single token*, the end-of-image (`[EOI]`). The `[EOI]` token accumulates visual semantics in early and mid layers and subsequently channels this information to textual tokens (section 3.2). Ablation (section 3.4) and activation-patching (section 4.1) experiments demonstrate that disrupting or modifying this token has a direct causal impact on model predictions, confirming that `[EOI]` serves as the central conduit for image-to-text communication.

In contrast, non-native multimodal models (e.g., LLaVA, Janus, Pixtral, VILA-U) display a more *distributed communication* pattern, in which visual semantics are shared across many image tokens. These results suggest that the emergence of localized, token-level communication is a specific property of native multimodal architectures and training regimes.

**The Role of Architecture in the Emergence of the Narrow Gate** The presence of a narrow gate is linked to the modality gap between image and text representations. Based on our results, we hypothesized that three architectural and training factors jointly influence its emergence. The **multimodal output objective**: a training objective that requires generating both image and text tokens, encourages the model to maintain distinct representational pathways for each modality, enforcing their separation. The **training history**: models trained *from scratch* on multimodal data, develop different subspaces for each modality, while models derived from pre-trained text-only LLMs and later fine-tuned for multimodal use (Janus, VILA-U) retain a text-aligned internal geometry and exhibit distributed communication. The **type of image encoder**: low-level visual tokenizers such as VQ-GAN (used in Chameleon), represent images through local features rather than high-level semantic embeddings. This increases the abstraction gap between image and text tokens and promotes the formation of a distinct communication interface centered on `[EOI]`. Taken together, these factors show that the narrow gate is not an incidental artifact but a structural consequence of jointly training a multimodal generator with low-level image tokenization.

**Implications and Future Directions** The localized communication via `[EOI]` introduces both limitations and opportunities. On one hand, the reduced communication bandwidth may constrain multimodal reasoning and expose the model to potential vulnerabilities. For example small perturbations or targeted manipulations of a single token can strongly influence output semantics. Encouraging a more distributed information flow, for instance by masking text-on-`[EOI]` attention during late-stage fine-tuning, could mitigate these issues. On the other hand, the narrow gate mechanism provides clear interpretability and practical benefits. It enables controlled steering of visual semantics, simplifies tracing how visual information shapes text generation, and facilitates efficient fine-tuning or alignment by localizing cross-modal transfer to a single token. Moreover, the `[EOI]` token acts as a natural *memory token* [32], summarizing global image information and allowing early image tokens to be discarded thereby reducing memory and computation costs [57, 58].

Overall, our study identifies the architectural and representational conditions under which narrow gate communication emerges in multimodal transformers. By bridging geometric analysis, causal interventions, and training design, our findings highlight how architectural choices determine whether visual–textual communication is concentrated through a single token or distributed across many, clarifying in this way an essential dimension of modern vision–language modeling.

# Limitations

Our analysis is limited to the image-to-text direction and to a subset of possible VLM architectures. We do not evaluate the communication in the text-to-image direction, or native VLMs that rely on higher-level, continuous encoding, and models with diffusion-based decoders. Additionally, our

study focuses only on native vision-language models. Extending these analyses to native multimodal models trained across a broader range of modalities remains an important open direction for future research.

## Acknowledgments

The authors acknowledge the AREA Science Park supercomputing platform ORFEO made available for conducting the research reported in this paper, and the technical support of the Laboratory of Data Engineering staff. Alessandro Pietro Serra, Francesco Ortu, Emanuele Panizon, Lorenzo Basile, Alessio Ansuini, Diego Doimo and Alberto Cazzaniga were supported by the project "Supporto alla diagnosi di malattie rare tramite l'intelligenza artificiale" CUP: F53C22001770002 and "Valutazione automatica delle immagini diagnostiche tramite l'intelligenza artificiale", CUP: F53C22001780002. Alessio Ansuini and Alberto Cazzaniga were supported by the European Union – NextGenerationEU within the project PNRR "PRP@CERIC" IR0000028 - Mission 4 Component 2 Investment 3.1 Action 3.1.1. Lucrezia Valeriani was supported by the project "QuB - Quantum Behavior in Biological Function" CUP: J95F21002820001. Lorenzo Basile was supported by the European Union – NextGenerationEU within the project PNRR "Finanziamento di progetti presentati da giovani ricercatori" - Mission 4 Component 2 Investment 1.2, CUP: J93C25000440001.

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

# A Related Works

**Special Tokens, Memory Tokens, Registers.** The importance of special tokens to store and redistribute global information was emphasized by Burtsev et al. [32]. In their work, they add these special tokens at the beginning of the sequence to serve as memory units accessible through the self-attention mechanism, improving performance on various language processing tasks. In vision transformers, Darcet et al. [57] found that high-norm tokens in low-informative areas of the images are used to store global information while discarding the local one. Adding learnable "register tokens" dedicated to global information processing helped eliminate the high-norm artifacts, improving performance. Similarly, Wen et al. [29] used learnable registers to summarize salient visual information and remove the image tokens altogether, improving the efficiency of vision-language models.

**Localization of information in Text-Only VLMs.** In text-only VLMs, this kind of summarization is particularly useful since much of the attention to image tokens is concentrated in the initial layers, likely on a few "anchor tokens" [28, 59]. Basu et al. [28] demonstrated with multimodal causal tracing that, in VQA tasks, a subset of strongly attended *late image tokens* in early layers transfer information to the text. However, subsequent studies have shown that important visual information is localized in heads of deeper layers of the network [60, 61], or within tokens corresponding to the spatial positions of objects in the image [27]. Interestingly, although these tokens have different statistical properties from those of textual tokens [62], they encode, in deeper layers, vocabulary words that describe the objects to which they correspond.

**Multimodal Vision Language Models.** Many approaches have been proposed to adapt decoder-only LLMs trained with autoregressive loss for image understanding [10–13, 63, 64] and multimodal generation tasks [16, 19–21, 24–26, 31, 34, 65, 66]. These models consist of an LLM, an image encoder, an image decoder (for multimodal-output VLMs), and adapters between the LLM and visual components. The success of diffusion-based image generation has led to the widespread use of stable diffusion as the decoding architecture in multimodal-output models [16, 19, 21, 24, 25, 66]. For the image encoding part, most architectures use a vision transformer (ViT), often that of CLIP [16, 19, 21, 24, 25, 34, 65, 66], due to the effectiveness of ViT embeddings in encoding image semantics [67]. The outputs from the visual encoder are either directly projected into the LLM embedding space [21, 24, 25, 67] or, in the case of multimodal generation, often mapped to a discrete set of tokens [16, 19, 65]. A drawback of these methods is the complexity of the encoding architecture. Moreover, the image-text contrastive loss can make the image representations resemble textual ones excessively [64], with text often favored as the preferred modality [17, 19, 25]. Alternative approaches use VQ-VAEs [68, 69] to encode and decode the images and train the LLMs from scratch on multimodal datasets [20, 26]. This *native multimodal* approach fuses the visual and textual representations early in training instead of using complex pipelines to bridge the modality gap [63, 67, 70, 71] and eliminates the need for adapters, greatly simplifying the architecture design.

# B Experimental Setup

## B.1 Models Architectures

Table A2 shows the detailed configurations of each model employed in the experiments.

| Parameters | Chameleon-7B | Chameleon-34B | Emu3-8B | LLaVA-7B | Pixtral-12B | Janus-1.3B | VILA-U-7B |
|---|---|---|---|---|---|---|---|
| N. Parameters | 7B | 34B | 8B | 7B | 12B | 1.3B | 7B |
| Backbone | - | - | - | Qwen2-7B | Mistral Nemo 12B | DeepSeek-LLM-1.3B | LLaMa-2-7B |
| Visual Encoder | VQ-GAN | VQ-GAN | VQ-GAN | SigLIP | Pixtral-ViT | VQ | RQ-VAE+SigLIP |
| Image Generation | Yes | Yes | Yes | No | No | Yes | Yes |
| N. Layers | 32 | 48 | 32 | 28 | 40 | 24 | 32 |
| N. Attention Heads | 32 | 48 | 32 | 28 | 32 | 16 | 32 |
| Hidden Size | 4096 | 8192 | 4096 | 3584 | 5120 | 2048 | 4096 |

Table A2: **Comparison of Model Architectures.** Detailed configurations of each model, including parameters, visual encoder type, generation capabilities, layer counts, attention heads, and hidden sizes.

## B.2 Datasets details

**Captioning prompt constructed from Flickr30K and MSCOCO.** The Flickr-30k [38] dataset and the MSCOCO dataset [39] contain respectively 30,000 and 330,000 images. Each sample of the first and approximately $200,000$ of the latter is associated with five captions. For the captioning benchmarks presented in tables 1 and A3 we constructed the prompts with the following structure: "⟨image⟩ `Provide a one-sentence caption for the provided image. __`"

**Image-text prompt constructed from Flickr-30k.** To generate prompts used in section 3.1 that alternate between text-first and image-first formats, we structured the inputs by concatenating all five captions for each image, either preceding or following the image itself, depending on the intended sequence.

**Question answering prompt constructed from VQAv2.** The dataset VQAv2 [37] consists of $204,721$ images from the MSCOCO dataset each associated with a variable number multiple choice question answering. For this benchmark presented in tables 1 and A3, we constructed prompts concatenating the image, the first of the associated question, and the instruction "`Answer the question using a single word or phrase.`"

**Image-text prompt constructed from ImageNet.** For the experiment described in sections 3.2 and 3.4 we use the following prompt construction "`<image> This animal is a`". For the activation patching experiment described in section 4.1 we use the same prompt for Chameleon-7B. For Chameleon-34B, we adopt a slightly modified prompt: "`<image> Answer the question using a single word, number, or short phrase. This animal is a`" as Chameleon-34B refused to respond to the original prompt.

For the activation patching experiment, described in section 4.1, we manually select 20 classes from the list above to ensure semantic diversity in the animal represented, and then we random sample 100 images for each class, obtaining a total of 2000 images. The selected pairs are:

- `(american_alligator.n.01, arabian_camel.n.01)`
- `(bald_eagle.n.01, barn_spider.n.01)`
- `(bee_eater.n.01, cheetah.n.01)`
- `(flamingo.n.01, great_grey_owl.n.01)`
- `(green_mamba.n.01, grey_whale.n.01)`
- `(hippopotamus.n.01, jaguar.n.01)`
- `(king_penguin.n.01, kit_fox.n.01)`
- `(lionfish.n.01, macaw.n.01)`
- `(proboscis_monkey.n.01, siberian_husky.n.01)`
- `(tailed_frog.n.01, trilobite.n.01)`

## B.3 Finetuning details

**Emu3-Gen finetuning.** We fine-tuned Emu3 on a mixture of datasets using 37.5k samples from the VQAv2 [37] and MS-COCO-2014 [39] training sets and 150k samples the LLaVA-instruct-150K using the Hugging Face implementation. We fine-tuned Emu3 for one epoch with a batch size of 64 using LoRA with a rank of 64, an alpha of 16, Adam optimizer without weight decay, and cosine annealing scheduler starting with a learning rate of 5e-5.

**Removing the narrow gate with fine-tuning.** We fine-tuned Chameleon-7b and Emu3 on a dataset of 15k samples, 7.5k samples from the LLaVA-instruct-150K training set, and 7.5k from the VQAv2 training set. We fine-tuned the models for one epoch with a batch size of 128, using LoRA with a rank of 128, alpha of 256, Adam optimizer with weight decay 0.1, cosine annealing learning scheduler with a starting learning rate of 2e-5.

### B.4 Advanced density peaks clustering.

The Advanced Density Peaks (ADP) clustering [52] is a mode-seeking, density-based clustering algorithm that finds the modes of the probability density on the data's low-dimensional data without performing any explicit dimensional reduction. We summarize the main steps below and refer the interested reader to the original paper [52].

The algorithm consists of three steps: the estimation of the data's intrinsic dimension, the estimation of the local density around each point, and a final density-based clustering of the data. Following previous works [72–74], we estimate the intrinsic dimension with `Gride`, a neighbor-based intrinsic dimension estimator, setting the rank $k$ of the nearest neighbor involved in the estimate to 16 (see [74] for more details). We then measure the local density around each data point with a $k$NN approach: $\rho_{i,k} = \frac{k}{NV_k}$. Here, $N$ is the number of data points, and $V$ is the volume of the ball, which has a radius equal to the distance between the point $i$ and its $k^{th}$ nearest neighbor. Importantly, we measure the volume on the intrinsic manifold using the intrinsic dimension value estimated in the first step.

The third step is the density-based clustering. With the knowledge of the $\rho_i$, we find a collection of density peaks $\mathcal{C} = \{c^1, ...c^n\}$, assign the data points around them, and find the density $\rho^{\alpha,\beta}$ of saddle points between a pair of clusters $c_\alpha$ $c_\beta$ with the procedure described in [52]. The statistical reliability of the peaks is assessed with a $t$-test on $\log \rho^\alpha - \log \rho^{\alpha,\beta}$, where $\rho^\alpha$ is the maximum density of peak $c_\alpha$, and $\rho^{\alpha,\beta}$ the density of the saddle point between $c_\alpha$ and $c_\beta$. Once the confidence level $Z$ is fixed, all the clusters that do not pass the $t$-test are merged since the value of their density peaks are considered indistinguishable from the nearby saddle point. The process is repeated until all the peaks satisfy the $t$-test and are statistically robust with a confidence $Z$ [52]. In the analysis reported in section 3.1, we remove clusters with size lower than 50.

## C   Additional Results

### C.1   Modality Gap in VLMs

We extend the results of section 3.1 to Janus, Pixtral, and VILA-U. As illustrated in Figure A1, all of the additional models exhibit an alignment between modalities that grows along the model depth, similar to the behavior observed in LLaVA. Figure A1-left confirms this alignment across modalities for all the models. Figure A1-right reveals that the homogeneity within modality clusters is lower compared to the Chameleon models and Emu3. This indicates that LLaVA, Janus, Pixtral, and VILA-U have a smaller modality gap.

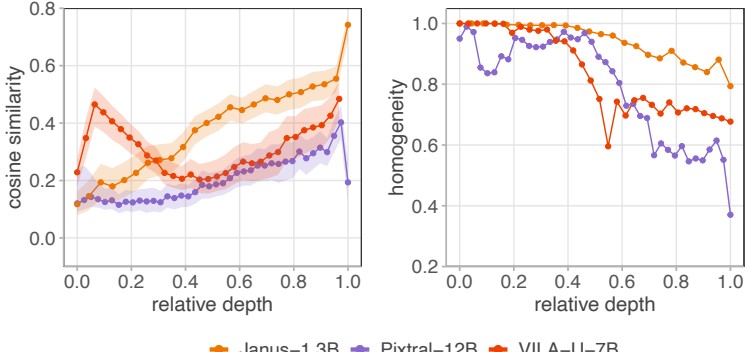

Figure A1: **Modality Gap in VLMs.** (**left**) Cosine similarity between text and image token embeddings as a function of model depth reflects the orthogonality of modalities in Janus, Pixtral, and VILA-U models. Points represent median cosine similarity, with shaded areas indicating the interquartile range. (**right**) Homogeneity score of token clusters generated via Advanced Density Peaks with respect to their original modality.

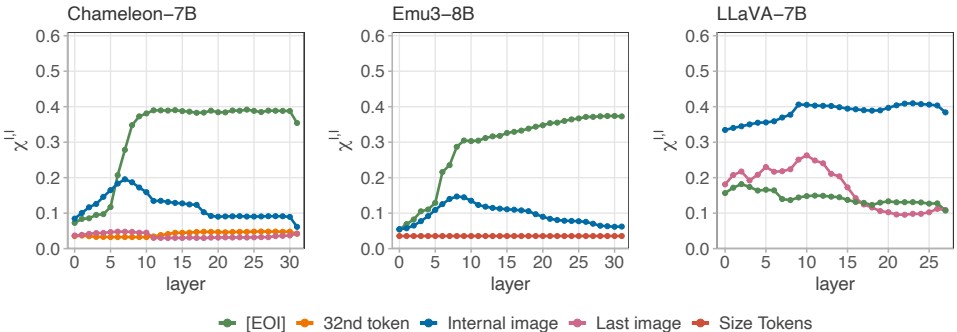

Figure A2: **Localization of Visual Semantic Information.** Neighborhood overlap between visual token embeddings and caption embeddings on the MS-COCO dataset for the models Chameleon-7B, Emu3-8B and LLaVA-7B. The results closely resemble those of figure 4

## C.2 A comparison of the semantic content of MS-COCO images in [EOI] and Qwen-GTE captions embeddings.

In section 3.3 we investigated the alignment between token representations across layers and the classes of ImageNet. In this section, we will now extend those findings to the MS-COCO captioning task to investigate if the results generalize to tasks different from classification. We will proceed as follows: first, for each image in the MS-COCO dataset, we extract the activations of the relevant visual tokens at each layer of every model analyzed in our study; Concurrently, we embedded the captions for these images using the Qwen2-7B-GTE text encoder [75]. To measure the alignment between these two sets of representations we used neighborhood overlap described at section 2.1, where in this context, the reference data representations are the embeddings from the text encoder. In other words for each sample $i$, we computed the fraction of shared $k$-nearest neighbors between the visual token embedding and the caption embedding according to equation (2). This approach allowed us to perform a direct comparison of the semantic alignment across all models and token types.

The results for the models of figure 4 are presented in figure A2, while the profiles for the remaining models are depicted in figure A3-right. We can notice that both group plots closely mirror the findings from our initial ImageNet experiments. This strong correspondence indicates that the localization of semantic information within specific tokens is a consistent characteristic of models like Chameleon and Emu, and that this phenomenon is not task-dependent. Furthermore, the high degree of alignment between the representations of these specialized visual tokens and the caption embeddings suggests that they encode high-level semantic abstractions that are comparable to those captured by sophisticated text-based sentence transformers.

## C.3 Cross-Modal Attention and Semantic Content of Visual Tokens

The results for Chameleon-34B closely mirror those of Chameleon-7B and Emu3, as discussed in section 3.2. Figure A3-left visualizes the cross-modal attention contributions of image tokens, where tokens contributing more than 1% individually are shown explicitly, while the rest are grouped as "internal image". In Chameleon-34B, cross-modal attention is primarily concentrated on three tokens—[EOI], the first image token, and the last. In contrast, Janus, Pixtral, and VILA-U exhibit a pattern similar to LLaVA (section 3.2), where attention to special tokens remains low. At the same time, a significant portion of the information is retained within internal image tokens. Figure A3-center further reveals that in Chameleon-34B as in the other *native multimodal* VLMs, [EOI] consistently maintains high $\chi_k^{l;gt}$ values across layers, indicating its dominant role in encoding visual information. Meanwhile, all other tokens exhibit near-zero overlap values, suggesting minimal contribution to meaningful visual representation. For Janus, both [EOI] and internal image tokens carry strong information from the earliest layers, implying that visual encoding may be directly influencing their representations. Similarly in Pixtral, both the [EOI] and the last image token exhibit a $\chi_k^{l;gt}$ higher than that of LLaVA, although still below Janus. However, as shown in appendix C.4, the information encoded in these special tokens do not play a significant role in information flow within the backbone models. Among the non-native multimodal VLMs, VILA-U most closely resembles LLaVA in terms

of semantic alignment, as most of the information is concentrated within the internal image tokens, while all other special tokens achieve negligible overlap scores.

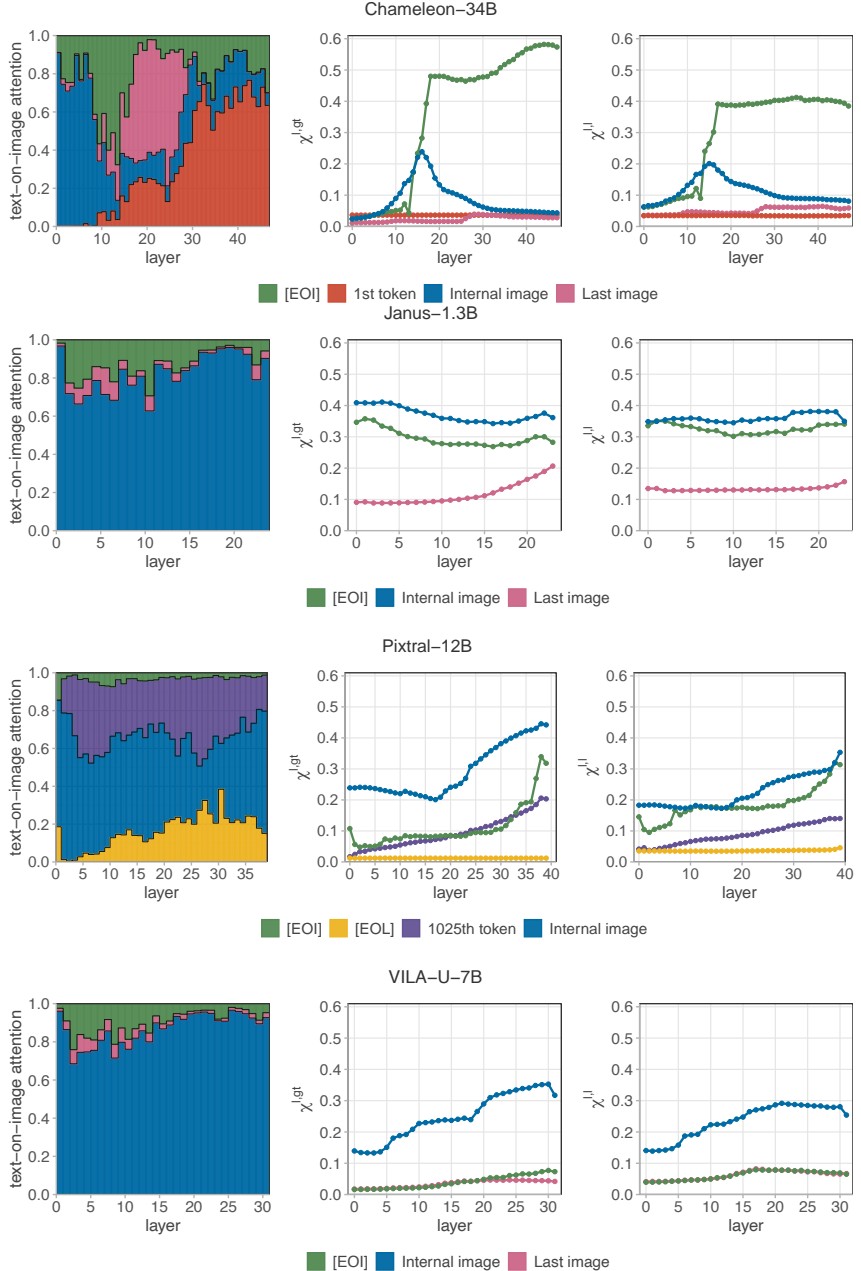

Figure A3: **Localization of Information in Different Models.** Each row corresponds to a model. The left panel shows the contribution of image token positions to total text-on-image attention. The center panel presents the neighborhood overlap between selected image tokens and ImageNet labels. The right panel finally presents the neighborhood overlap between visual token embeddings and caption embeddings on the MS-COCO dataset.

## C.4 Effect of attention knockout on vision-language tasks

Table A3 extends the results of table 1, showing the ablation effect on all tokens identified for their strong cross-modal attention. The analysis further confirms that none of the special tokens, except for [EOI], play a significant role, as their ablation does not noticeably affect performance on any benchmarks.

| Model | Ablation | VQAv2 | Flickr | MS-COCO | ImageNet ($\chi_k^{out,gt}$) |
|---|---|---|---|---|---|
| Chameleon-7B | - | 0.51 | 0.34 | 0.48 | 0.46 |
| | text $\rightarrow$ [EOI] | **0.25** | **0.04** | **0.04** | **0.01** |
| | text $\rightarrow$ Last-Image | 0.42 | 0.20 | 0.32 | 0.46 |
| | text $\rightarrow 32^{nd}$ | 0.51 | 0.34 | 0.49 | 0.46 |
| Chameleon-34B | - | 0.59 | 0.41 | 0.47 | 0.43 |
| | text $\rightarrow$ [EOI] | **0.39** | **0.01** | **0.01** | **0.04** |
| | text $\rightarrow$ Last-Image | 0.58 | 0.39 | 0.46 | 0.41 |
| | text $\rightarrow$ First-Image | 0.57 | 0.38 | 0.46 | 0.42 |
| Emu3 | - | 0.57 | 0.29 | 0.63 | 0.35 |
| | text $\rightarrow$ [EOI] | 0.48 | **0.13** | **0.32** | **0.24** |
| | text $\rightarrow$ img | **0.42** | 0.22 | 0.54 | 0.30 |
| LLaVA | - | 0.8 | 0.70 | 0.98 | 0.5 |
| | text $\rightarrow$ [EOI] | 0.8 | 0.71 | 0.97 | 0.45 |
| | text $\rightarrow$ img | **0.00** | **0.00** | **0.01** | **0.05** |
| Pixtral | - | 0.79 | 0.59 | 0.68 | 0.63 |
| | text $\rightarrow$ [EOI] | 0.77 | 0.62 | 0.68 | 0.61 |
| | text $\rightarrow$ [EOL]s | 0.76 | 0.54 | 0.61 | 0.61 |
| | text $\rightarrow$ Last-Image | 0.77 | 0.57 | 0.68 | 0.62 |
| | text $\rightarrow 1025^{th}$ | 0.77 | 0.55 | 0.62 | 0.61 |
| | text $\rightarrow$ img | **0.37** | **0.00** | **0.01** | **0.06** |
| Janus | - | 0.51 | 0.32 | 0.43 | 0.65 |
| | text $\rightarrow$ [EOI] | 0.51 | 0.32 | 0.42 | 0.65 |
| | text $\rightarrow$ img | **0.00** | **0.01** | **0.01** | **0.06** |
| VILA-U | - | 0.71 | 0.42 | 0.61 | 0.64 |
| | text $\rightarrow$ [EOI] | 0.72 | 0.43 | 0.61 | 0.65 |
| | text $\rightarrow$ Last-Image | 0.72 | 0.42 | 0.61 | 0.64 |
| | text $\rightarrow$ img | **0.39** | **0.01** | **0.02** | **0.08** |

Table A3: **Effect of Attention Knockout on Image Understanding Tasks**. Performance of the models on visual question answering (VQAv2), image captioning (Flickr-30k and MS-COCO), and image classification (ImageNet) under different ablation settings. We ablated the special tokens with high cross-modality attention by removing their communication with the text tokens. Numbers in bold mark the worst performance for each model and task.

