# OpenReview forum: "The Narrow Gate: Localized Image-Text Communication in Native Multimodal Models"
_NeurIPS.cc/2025/Conference — NeurIPS 2025 poster_

### Official Review · Reviewer_gNmN · 2025-06-20

**Clarity:** 4
**Significance:** 2
**Originality:** 2
**Rating:** 5
**Confidence:** 4

**Summary:**

In this submission, the authors perform a detailed experimental evaluation regarding the “communication mechanism” between different modalities (i.e., image and text) in pre-trained Vision-Language-Models (VLMs). Specifically, in their evaluation they compare text-only output models with models that are able to generate multimodal outputs. The authors argue that there is a clear difference between these two types of models when it comes to how information from the image modality flows to the textual modality. Specifically, they find that models trained to only produce textual outputs have more distributed attention patterns than those that are also able to generate images, which the authors find to mainly communicate through a “narrow gate” (the end-of-image EOI) token.

The authors provide a broad evaluation showcasing a difference in behavior between model types in general and with respect to the narrow gate in particular, including similarity metrics in embedding space, patching interventions, clustering analysis, and attention knockout experiments.

**Questions:**

Please see the above weaknesses. In particular, I would highly appreciate if the authors could elaborate on the significance of their findings. Additionally, I would be curious to hear the authors' opinion on the stated concerns regarding the interpretation of the results.

**Ethical Concerns:**

["NO or VERY MINOR ethics concerns only"]

**Final Justification:**

Overall, I think this submission discusses an interesting phenomenon that is quite relevant given the current interest in multi-modal large language models the community. While I agree with reviewer 9UB8 that the _utility_ of this finding is not fully clear, the finding in itself is interesting and the experiments well-executed. I view this as an interesting addition in the context of explainable and interpretable AI, in the sense that it sheds light on the inner workings of highly relevant models. I therefore raise my score to accept.

**Limitations:**

yes

**Quality:**

3

**Strengths And Weaknesses:**

**Strenghts.**

- First of all, I would like to commend the authors on the general quality of the submission — the information is well-presented, the writing and structure are clear, and the analyses and conclusions well presented.
- The authors carefully design a multitude of experiments to showcase the behavioral differences between the models, employing adequate and state-of-the-art techniques for doing so. This allows them to distill a clear mechanistic difference between model types that can be tested through interventions.
- While the analysis is quite *narrow* (focusing on a highly specific mechanism in the model, the *narrow gate*), the in-depth analysis and experimental evaluation yields reliable results for future research to build on.

**Weaknesses.**

While the evaluation and presentation of the observed effect are convincing, I have a couple of remaining questions and concerns.

- **Relevance.** While the authors results convince me that there is a difference between models additionally trained for image generation and those that only produce text, the practical relevance of this finding is still a bit unclear to me. This is in particular the case as the results are also not overly surprising — if there is a loss on the image tokens to produce a very specific kind of information, it seems to be expected that these tokens will take a different role compared to when no loss is applied to the image tokens. Specifically, with no loss on the image tokens, there is no pressure on the EOI token to capture much semantics about the image, as the image tokens themselves can directly be used. Why is this finding of importance?
- **Interpretation of results.**
    - The authors argue that the results show that “the semantic content is more distributed [… and the models rely …] on multiple regions of the image” rather than a single token. It is unclear to me if, given the current results, much of a conclusion about the representation of information in the image tokens can be drawn (such as them representing different image regions in a specific fashion). Specifically, it seems to me that a very simple hypothesis could explain most of the respective results: the image tokens might just have a lot of redundant information (similar embeddings), which would be consistent with their pretraining paradigm, in which the spatial dimensions are averaged over to yield a single semantic embedding vector. As a result, deleting any single token or even larger subsets of tokens is unlikely to make a difference — importantly, highly similar embeddings would also induce the observed highly distributed attention maps. I would be curious to know the authors' take on this; image token deletion experiments or differently trained embedding models (e.g. DINO) could help better understand the underlying reason for the observed attention patterns.
    - Secondly, I have difficulties with the argument that in Llava-7B (or other text-only models) the “image embeddings and text embeddings align increasingly throughout the layers”. Specifically, what does it mean to be an image embedding later in the network? As no loss is applied to the output at the position of image tokens, assigning any modality label to these tokens seems arbitrary. The model might just use the compute of those additional, potentially redundant tokens to improve performance (see also register tokens (Darcet et al, 2023) or pause tokens (Goyal et al., 2023).

Additional minor weaknesses.

- Figure 1 (right) is cumbersome to read, as 3 dimensions are shown in 2 dimensions without the 3rd axis being clear.
- The authors evaluate a custom version of Emu3 (fine-tuned from Emu3-Gen) — why is this relevant and how do results differ between models?
- Minor typos: line 133 fig 2 is lowercase at the beginning of the sentence, in line 150 the word appendix is repeated, in line 169 there is a missing punctuation.

---

> ### Author Rebuttal · Authors · 2025-07-31
>
> Thank you for your thoughtful and detailed review. We appreciate the time and care you took in evaluating our work, as well as your recognition of its key strengths. We address your main concerns point by point below.
>
> **Relevance**
> > *The practical relevance of this finding is still a bit unclear to me.*
>
> Between lines 323 and 333, we discuss several potential practical implications of our findings, showing applications where the narrow gate can be exploited proactively (e.g., steering via patching) and propose how to use it to reduce memory and computing usage. \
> We also suggest that a possible way to make the models more robust and “*encourage a more distributed communication can be masking the text-on-[EOI] attention during the last part of training*” (lines 325-326).
> To validate experimentally the last claim and address the reviewer's concern, we fine-tuned Chameleon and Emu3, masking the [EOI] tokens on an instruction dataset composed of 10k samples from the LLaVA training set and 5k samples from the VQAv2 training set. In this case, we can significantly reduce the detrimental effect of [EOI] ablation on performance.
> The results evaluated on 50 validation samples each from VQAv2 and COCO are summarized below:
>
> - *Chameleon*:
>     - VQAv2 accuracy improved from 50% to 62% (base). With ablation of [EOI], it improves from 0.05% to 52%.
>     - COCO-CIDER score increased from 0.48 to 0.72 (base). With ablation of [EOI], the score increased from 0.06 to 0.66.
>
> - *Emu3*:
>     - VQAv2 accuracy improved from 44 to 50% (base). With ablation of [EOI], it improves from 35% to 46%.
>     - COCO-CIDER score increased from 0.73 to 0.87 (base). With ablation of [EOI], the score increased from 0.40 to 0.82.
>
> To validate the importance of keeping [EOI] masked during training, we performed a standard fine-tuning run without masking [EOI]. Then, we repeat the ablation experiment in Tab. I and we measured the effect of ablation [EOI] after the fine-tuning.
> - *Chameleon*: Final VQAv2 accuracy was 53% (base) and 35% with [EOI] ablation. COCO-CIDER was 0.75
>  (base) and 0.05 with [EOI] ablation.
>
> - *Emu3*: Final VQAv2 accuracy was 45% (base) and 43% with [EOI] ablation and COCO-CIDER was 0.82 (base)  and 0.48 with [EOI] ablation.
>
> These results show that the training with [EOI] masked is an effective strategy to remove almost completely the gating mechanism of [EOI]. Notably, masked fine-tuning also shows improved performance wrt the non-masked standard, except for COCO-CIDER in Chameleon.
> The masked-finetuning, targeted on [EOI], is an example of how the findings of this work can be used to improve the model's performance and robustness.
>
> > *This is in particular the case as the results are also not overly surprising [...] Specifically, with no loss on the image tokens, there is no pressure on the EOI token to capture much semantics about the image, as the image tokens themselves can directly be used. Why is this finding of importance?*
>
> Through our experiments, we show that having multimodal output, i.e., having a loss on the image tokens, is not sufficient to present a narrow gate: the Janus model (and the Liquid model we have integrated during the rebuttal phase) are multimodal in output, but they do not present a narrow gate. Our results not only show the (hitherto unreported) existence of a multimodal narrow gate, but also shed light on its possible causes.
>
> The fact that native multimodal-output models, as Chameleon and Emu3, construct an efficient communication strategy despite exhibiting a wide modality gap across all the layers (Fig. 3) is rather surprising. Indeed, the modality gap can create challenges for performance in multimodal tasks [1] and can be used to quantify the quality of the training in VLMs [2]. Understanding and characterizing how native multimodal-output models develop an effective bottleneck-style communication strategy is an important insight for future studies in multimodal models, especially given their recent implementation in several state-of-the-art multimodal training pipelines [3,4].
>
>
> **Interpretation of results.**
> >  *The authors argue that the results show that...*
>
> The reviewer offers an interesting interpretation of why semantic content appears diffuse in models that do not exhibit a narrow-gating mechanism (e.g., Llava and Pixtral).
>
> However, our work specifically focuses on models like Chameleon and Emu3, which show the opposite behavior, where semantics are localized in a single token (the [EOI] token). In these models, we observe a significant embedding gap between image and text modalities throughout the hidden layers (see Fig. 1), much larger than in models where the semantic content is distributed, which suggests that this [EOI] may function as a modality-bridging messenger token.
> This hypothesis can be motivated by the very low cosine similarity between [EOI] with both text and image embeddings (<0.1), indicating it doesn't belong to either modality. Instead, it seems to act as an intermediary to enable cross-modal communication.
> The structural difference, diffuse versus localized communication, might stem from how the models handle the alignment between modalities, and we agree that further experiments (e.g., token ablations or alternative pretraining approaches like DINO) could provide deeper insight.
>
> >  *Secondly, I have difficulties with the argument that in Llava-7B (or other text-only models) the “image embeddings and text embeddings align increasingly throughout the layers”*
>
> Our definition of image embedding is adopted following the convention in the literature [e.g. 5]. We simply refer to the embeddings of the tokens from the image.
> Moreover, previous work on the Llava-style models [5] showed that  “*activations in the late layers at each visual token position correspond to token embeddings that describe its original patch object in its corresponding image token*”. Image embeddings at late layers, therefore, do preserve interpretable visual information of the input, even when no loss is applied to those positions' output.
>
>
> ### Additional minor weaknesses.
>
> > *Figure 1 (right)  is cumbersome to read, as 3 dimensions are shown in 2 dimensions without the 3rd axis being clear.*
>
> We are not sure we understand the reviewer’s comment.  Figure 1 (right) shows how different models perform after ablation of the two possible communication channels: image-text (x-axis) and [eoi]-text (y-axis). There isn’t a third axis; the colors of the markers simply denote different models.
>
> > *The authors evaluate a custom version of Emu3 (fine-tuned from Emu3-Gen) — why is this relevant and how do results differ between models?*
>
> We fine-tuned the Emu3-Gen model since we wanted a model capable of both generating images and text.
>
> > *Minor typos [...]*
>
> We thank the reviewer. These typos will be corrected in the final version.
>
>
> [1] W. Lieng et al., Mind the gap: Understanding the modality gap in multi-modal contrastive representation learning. NeurIPS, 2022
>
> [2]  Q. Huang et al., Deciphering Cross-Modal Alignment in Large Vision-Language Models with Modality Integration Rate, ICCV, 2025
>
> [3] openai.com/index/introducing-4o-image-generation/
>
> [4] M. Shukor et al., Scaling Laws for Native Multimodal Models, 2025, arXiv:2504.07951
>
> [5] Neo et al. Towards Interpreting Visual Information Processing in Vision-Language Models.
>
> ---
>
> Thank you again for your comments and suggestions. If we manage to address your main concerns, we politely ask you to consider raising your score. Please don’t hesitate to let us know if you have further questions. We are happy to answer your questions during this rebuttal discussion period.

---

> > ### Comment · Reviewer_gNmN · 2025-08-03
> >
> > I would like to thank the authors for their detailed response to my questions, which addresses my concerns; further, I believe the authors' answer to the concerns raised by 9UB8 to address those concerns as well. I will gladly increase my score. Minor additional comments below.
> >
> > >  To validate experimentally the last claim and address the reviewer's concern, we fine-tuned Chameleon and Emu3, masking the [EOI] tokens on an instruction dataset composed of 10k samples from the LLaVA training set and 5k samples from the VQAv2 training set. In this case, we can significantly reduce the detrimental effect of [EOI] ablation on performance.
> > The results evaluated on 50 validation samples each from VQAv2 and COCO are summarized below:
> > Chameleon:
> > VQAv2 accuracy improved from 50% to 62% (base). With ablation of [EOI], it improves from 0.05% to 52%.
> > COCO-CIDER score increased from 0.48 to 0.72 (base). With ablation of [EOI], the score increased from 0.06 to 0.66.
> > Emu3:
> > VQAv2 accuracy improved from 44 to 50% (base). With ablation of [EOI], it improves from 35% to 46%.
> > COCO-CIDER score increased from 0.73 to 0.87 (base). With ablation of [EOI], the score increased from 0.40 to 0.82.
> > To validate the importance of keeping [EOI] masked during training, we performed a standard fine-tuning run without masking [EOI]. Then, we repeat the ablation experiment in Tab. I and we measured the effect of ablation [EOI] after the fine-tuning.
> > Chameleon: Final VQAv2 accuracy was 53% (base) and 35% with [EOI] ablation. COCO-CIDER was 0.75 (base) and 0.05 with [EOI] ablation.
> > Emu3: Final VQAv2 accuracy was 45% (base) and 43% with [EOI] ablation and COCO-CIDER was 0.82 (base) and 0.48 with [EOI] ablation.
> > These results show that the training with [EOI] masked is an effective strategy to remove almost completely the gating mechanism of [EOI]. Notably, masked fine-tuning also shows improved performance wrt the non-masked standard, except for COCO-CIDER in Chameleon. The masked-finetuning, targeted on [EOI], is an example of how the findings of this work can be used to improve the model's performance and robustness.
> >
> > These results are indeed very interesting and provide additional evidence for the authors' interpretation of their results. I would encourage the authors to add them to the final manuscript.
> >
> > > Through our experiments, we show that having multimodal output, i.e., having a loss on the image tokens, is not sufficient to present a narrow gate: the Janus model (and the Liquid model we have integrated during the rebuttal phase) are multimodal in output, but they do not present a narrow gate. Our results not only show the (hitherto unreported) existence of a multimodal narrow gate, but also shed light on its possible causes.
> >
> > Indeed, sorry I missed that during my initial review.
> >
> >
> > > We are not sure we understand the reviewer’s comment. Figure 1 (right) shows how different models perform after ablation of the two possible communication channels: image-text (x-axis) and [eoi]-text (y-axis). There isn’t a third axis; the colors of the markers simply denote different models.
> >
> > I still believe a table would make the results much easier to parse. What I am referring to is the fact that the labels of the axes ('No [EOI] attention' and 'No image text attention') do not correspond to the variable that changes along the axes, which is relative performance in both cases.

---

> > > ### Author Response · Authors · 2025-08-04
> > >
> > > > *These results are indeed very interesting and provide additional evidence for the authors' interpretation of their results. I would encourage the authors to add them to the final manuscript.*
> > >
> > > Thank you for your appreciation of our additional results. We will add the fine-tuning experiment at the end of the result section (Sec. 3.5)  in the camera-ready version of this work.
> > >
> > > >  *Still believe a table would make the results much easier to parse.*
> > >
> > > We now understand the source of confusion. In the final version of this work, we will either update the labels of the axes, clarifying which variable they refer to (for instance, using “relative performance without [EOI] attention”, ...), or use a table as you suggested instead.
> > >
> > > > *I would like to thank the authors for their detailed response to my questions, which addresses my concerns; further, I believe the authors' answer to the concerns raised by 9UB8 to address those concerns as well. I will gladly increase my score.*
> > >
> > > We deeply thank you for your strong endorsement and the recognition that your concerns have been positively addressed.

---

### Official Review · Reviewer_mkQ5 · 2025-06-25

**Clarity:** 3
**Significance:** 3
**Originality:** 2
**Rating:** 4
**Confidence:** 4

**Summary:**

This paper investigates how visual information is transferred to textual outputs in vision-language models (VLMs), with a specific focus on the internal communication patterns between image and text tokens. The authors compare multimodal-output VLMs (e.g., Chameleon, Emu3) with unimodal-output VLMs (e.g., LLaVA, Pixtral), identifying a key difference: the former tend to use a single “narrow gate” token—EOI(end-of-image)—as the main conduit for visual information, while the latter employ distributed communication across many image tokens. Through a combination of attention analysis, semantic probing, ablation studies, and activation patching, the authors demonstrate that in narrow-gate models, blocking or modifying the EOI token drastically alters performance and output semantics. This highlights the architectural and functional distinction between these classes of models.

**Questions:**

see weakness.

**Ethical Concerns:**

["NO or VERY MINOR ethics concerns only"]

**Final Justification:**

My issues were resolved, I agree raising my score.

**Limitations:**

yes

**Quality:**

2

**Strengths And Weaknesses:**

Strengths

1. The paper provides a detailed investigation of cross-modal communication mechanisms inside VLMs, using diverse interpretability tools such as attention knockout, cosine similarity, semantic probing (χ metric), and activation patching.

2. The discovery that multimodal-output VLMs rely on a single token for image-to-text information transfer has broad implications for model control, interpretability, and robustness.

3. The analysis is conducted across several strong open-source VLMs (Chameleon, Emu3, LLaVA, Janus, Pixtral), and the conclusions are supported by consistent evidence across different models and tasks (VQA, captioning, classification).

Weaknesses

1. The work is primarily an empirical analysis rather than a method or model contribution. While the findings are meaningful, the lack of algorithmic novelty may limit its perceived impact in NeurIPS.


2. The paper omits discussion of prior studies[1,2,3,4,5,6] on communication patterns between image and text tokens, some findings are already observed in prior work.

[1] Mitigating Multimodal Hallucination from an EOS Decision Perspective.

[2] Mitigating Object Hallucination via Concentric Causal Attention.

[3] Alleviating Hallucination in Multi-Modal Large Language Models via Over-Trust Penalty and Retrospection-Allocation.

[4] Controlmllm: Training-free visual prompt learning for multimodal large language models.

[5] Attention Reallocation: Towards Zero-cost and Controllable Hallucination Mitigation of MLLMs.

[6] An image is worth 1/2 tokens after layer 2: Plug-and-play inference acceleration for large vision-language models.


3. The paper studies only limited models. It is unclear whether the phenomenon holds in other models (e.g. Qwen2.5-VL, InternVL2.5). Broader generalization is needed for the findings to have lasting significance.

4. The paper appears somewhat rushed and underdeveloped in its experimental design. Several claims are made based on limited observations, without the breadth of analysis necessary to establish them as robust conclusions.

---

> ### Author Rebuttal · Authors · 2025-07-31
>
> Thank you for your feedback and the analysis of our manuscript. We will address below the concerns you raised.
>
> ### Weaknesses
>
> > *1. The work is primarily an empirical analysis...*
>
> We respectfully disagree with the characterization that empirical contributions lack value at NeurIPS. The NeurIPS call for papers explicitly states: *'We also encourage in-depth analysis of existing methods that provide new insights in terms of their limitations or behaviour beyond the scope of the original work."*  While our contribution is empirical, we address a well-recognized problem of characterizing the differences in information flow across modalities in VLMs. By exposing a single-token bottleneck that can both steer and break multimodal models, our study provides actionable knowledge for robustness, safety, and efficiency research.
> We believe empirical contributions are fundamental to scientific progress and provide essential foundations for future algorithmic developments.
>
> > *2. The paper omits discussion of prior studies...*
>
> We thank the reviewer for the suggestions. We will add the relevant literature in the related work section. However, all of these papers deal with VLMs that output only text. The main finding of our paper is about VLMs, which can perform jointly in image understanding and image generation tasks and, as such, remain entirely novel.
>
> > *3. The paper studies only limited models*
>
> In our work, we explore the cross-modal communication in a diverse set of models. We study i) VLMs which output images and text (Emu3, Chameleon, Janus) and output only text (Pixtral, Llava)  ii) VLMs trained from scratch (Emu3, Chameleon) and fine-tuned on pre-trained LLM backbones (Pixtral, Llava, Janus), and iii) models with discrete, low-level VQ-VAE vision encoders (Chameleon, Emu3) and high-level, text-aligned CLIP-like encoders (Janus, Pixtral, Llava).
> Both Qwen2.5-VL and InternVL2.5 are multimodal-in-input but unimodal in output, like Llava and Pixtral. As we write in lines 54-55 and 319-321, our findings show that the narrow gate allows for selective ablation and steering, but *“none of these properties hold for Llava and other unimodal-output VLMs”*, and the ability to *“produce multimodal responses is crucial”* for the emergence of this phenomenon. For these reasons, based on the experiments we performed on LLava and Pixtral, we do not expect these models to present a narrow gate.
>
> > *4. The paper appears somewhat rushed and underdeveloped...*
>
> The reviewer previously remarked that *“the conclusions are supported by consistent evidence across different models and tasks”* and this point of view is also shared by reviewer R1ay, who recognises that we base our claims on extensive statistics across several VLMs.
> Our conclusions stem organically from the experimental results. 1) The high attention weight and large semantic content indicate [EOI] to be a possible narrow gate for Chameleon and Emu, and 2) the selective ablation proves this to be correct, since the removal of a single token effectively destroys performance over a diverse set of downstream tasks.
>
> ---
> If the reviewer has more specific concerns about any of these points, we would be happy to provide clarification in the author-reviewers discussion phase. Otherwise, we politely ask them to consider raising their score.

---

> > ### Author Response · Authors · 2025-08-05
> >
> > As the discussion period is coming to an end, we kindly ask you whether your concerns have been addressed by our responses.
> >
> > We thank you again for the time spent reviewing our paper.

---

> > ### Comment · Area_Chair_89jf · 2025-08-05
> > **Requesting Response to Rebuttal**
> >
> > Dear Reviewer mkQ5,
> >
> > please respond to the authors, so that the authors know their rebuttal has been read.
> >
> > Best regards, AC

---

> ### Comment · Reviewer_mkQ5 · 2025-08-06
>
> The authors' response is too brief and does not provide useful information. In particular, some important conclusions in the paper have already been presented in the works I cited, and the authors have not included the additional experimental results that I consider necessary. However, after seeing the discussions from other reviewers, I believe there is no longer a need for me to participate in the discussion. If the authors clearly explain how they will address the relationship with strongly related prior work in the final version, I will consider raising my score.

---

> ### Author Response · Authors · 2025-08-07
>
> We thank the reviewer for their reply. Below, we address the remaining concerns:
> > The authors' response is too brief and does not include the experiments I requested.
>
> While we did not run experiments on Qwen2.5-VL and InternVL2.5, we did conduct experiments on models that belong to the same class, and our findings are relevant for the models suggested by the reviewer. Specifically, both Qwen2.5-VL and InternVL2.5 share the following characteristics:
> 1. **Training**: both are optimized with a next-token language-modeling loss on their text streams.
> 2.  **Architecture**: each builds on a large, pre-trained transformer-based language model backbone and a MLP connector.
> 3.  **Visual encoders**: all employ a Vision Transformer (ViT) to extract image features.
>
> All of these elements can be found in Pixtral-12b and LLaVA-Onevision, which we experimentally analyze in our work and show no evidence of a narrow gate. As we discuss, the necessary conditions for the narrow gate phenomenon are: a) models capable of generating multimodal output, b) models trained from scratch on multimodal data, and c) models with a low-level vision encoder. None of these three criteria are met in Qwen2.5-VL or InternVL2.5.
>
> Thus, while we did not directly experiment on Qwen-2.5-VL and InternVL2.5, our analyses on closely related models provide strong evidence that the narrow gate phenomenon does not occur in this class of models.
>
> >  If the authors clearly explain how they will address the relationship with strongly related prior work in the final version, I will consider raising my score.
>
>  We briefly expand on the merits of these works in connection with our findings:
>
>  1. **Mitigating Multimodal Hallucination from an EOS Decision Perspective**.
> This work studies information flow in LLaVA on captioning tasks, observing the distributed integration of visual and textual information when predicting EOS. This work does not directly probe the semantic content or localization of information in EOS, nor its role in cross-modal communication, a key question in our study.
>
>  2. **Mitigating Object Hallucination via Concentric Causal Attention**.
> It analyzes custom LLaVA-style models and shows that the information flow from distant visual tokens is weaker due to a decaying effect in the RoPE positional embedding, causing hallucination when the objects are far away from the referring textual part of the prompt.
> We discussed a related phenomenon in lines 314 (Discussion) and 224-225 (Appendix). We will explicitly discuss this work in our revised paper.
>
>  3. **Alleviating Hallucination in Multi-Modal Large Language Models via Over-Trust Penalty and Retrospection-Allocation.**
> It shows that in text-only output VLMs, highly attended textual tokens of little semantic information tend to aggregate the content of a previous sentence,  similar to register tokens ([a,b] already cited). We will expand our discussion of this connection in the Appendix section *“Special Tokens, Memory Tokens, Registers.”*
>
> 4. **Controlmllm: Training-free visual prompt learning for multimodal large language models** proposes a post-hoc method for enhancing text-to-image attention. This is distinct from our work, as the narrow gate phenomenon we describe arises naturally in model architecture and training, not via inference-time interventions.
>
> 5. **Attention Reallocation: Towards Zero-cost and Controllable Hallucination Mitigation of MLLMs** is very similar to 3. We also note that this work is concurrent with our submission accordign to the NeurIPS 2025 policy (see [c]). We will include this work in the same place as 3.
>
> 6. **An image is worth 1/2 tokens after layer 2**.
> We have already discussed this work in our section on *“Information Flow in Text-Only VLMs and Special Tokens”* in the related work (Appendix). However, since reviewers are not required to read the appendix in detail, we understand it may not have been visible. In the final version, we will additionally reference it at the end of our Discussion (l. 330–333) when connecting to memory tokens and computational efficiency strategies.
>
> ---
>
> We hope these clarifications address your concerns.
>
> ### References
> [a] Darcet et al., Vision Transformers Need Registers.
>
> [b] Neo et al., Towards Interpreting Visual Information Processing in Vision-Language Models
>
> [c] neurips.cc/Conferences/2025/PaperInformation/NeurIPS-FAQ

---

> > ### Comment · Area_Chair_89jf · 2025-08-08
> >
> > Dear reviewer mkQ5,
> >
> > it would be interesting to hear whether the authors' reply (in particular regarding "address the relationship with strongly related prior work") address some of your questions or concerns?
> >
> > Thank you & best regards, AC

---

### Official Review · Reviewer_R1ay · 2025-06-25

**Clarity:** 3
**Significance:** 3
**Originality:** 2
**Rating:** 4
**Confidence:** 3

**Summary:**

This paper investigates how visual information is processed and transferred to the textual domain by several representative multimodal-output and unimodal-output VLMs. The work finds that multimodal-output VLMs more rely on a single token for visual information while unimodel-output VLMs exhibit a distributed communication pattern. Modifying the narrow gate token enables steering of the image semantics to control the output behavior.

**Questions:**

My main concerns are items 1–3 under Weaknesses. If these are addressed, I will consider raising the rating.

**Ethical Concerns:**

["NO or VERY MINOR ethics concerns only"]

**Final Justification:**

This paper reveals the "narrow gate" phenomenon in the image-text transmission process of multimodal large models through extensive experiments. My primary concern lies in the practical significance of this finding; however, given that the authors provided a downstream application case by masking attentions to the narrow gate token when training during the rebuttal stage, I decide to raise the rating from 3 to 4.

**Limitations:**

yes

**Quality:**

3

**Strengths And Weaknesses:**

Strengths:
1. This paper is well-written. It introduce the findings of narrow gate token for visual information in multimodal-output VLMs and distributed communication pattern in unimodel-output VLMs by extensive statistics across several VLMs.
2. The work presents that editing narrow gate token can control the global visual information.

Weaknesses:
1. The conclusions presented in the Abstract and Introduction seem to conflict with those in Section 4. According to Figure 1, Janus stands out as an outlier among multimodal-output VLMs. It does not exhibit the narrow gate token pattern but instead aligns more closely with the distributed communication pattern typical of unimodal-output VLMs. Therefore, Section 4 concludes that the presence of narrow gate tokens depends on whether the model is jointly trained from scratch on vision and language data. However, at the beginning of the paper, the classification is based solely on the output modality—multimodal versus unimodal. I suggest revising the conclusions in the Abstract and Introduction to maintain consistency. Furthermore, given the substantial differences among models, this conclusion is difficult to verify. Communication patterns may be influenced not only by the training strategy (i.e., joint vision-language pretraining) but also by the architecture of the vision encoder.
2. While I appreciate the paper’s careful and well-reasoned investigation, the contributions are somewhat difficult to assess. The narrow gate token, also known as a memory or pivot token, is a widely observed phenomenon in both LLMs and VLMs. After identifying the narrow gate token pattern, I believe the paper should go further by proposing ideas to mitigate the drawbacks of this pattern and support them with experimental validation—even small-scale experiments would be valuable. For example, in Section 4, the authors suggest that “a possible approach to encourage a more distributed communication can be masking the text-on-[EOI] attention during the last part of training,” but this idea lacks empirical support.
3. The experiments in the paper are limited to the image classification task, namely: “⟨image⟩ This animal is a __.” I believe it is important to expand the exploration to include broader question-answering scenarios to enhance the generalizability of the findings.
4. The paper focuses exclusively on the information flow from vision to text. Considering that the multimodal-output VLMs are introduced, I am curious whether the narrow gate token pattern also emerges during image generation tasks in multimodal-output VLMs. This would relate to the vision-text-to-vision communication paradigm, which warrants further exploration.

---

> ### Author Rebuttal · Authors · 2025-07-31
>
> Thank you for your valuable feedback and the thorough analysis of our manuscript. We will use your suggestions to improve the quality of our work. We address the concerns you raised below.
>
> ### Weaknesses
> 1. > The conclusions presented in the Abstract and Introduction seem to conflict with those in Section 4. [...]
>
> We agree with the reviewer. We will modify the Abstract and Introduction to be more consistent with our discussion and conclusions. Indeed, there is an impact of the diversity of the architectures (visual encoder) and training protocols (training from scratch vs fine-tuning of the LLM backbone) on the narrow gate phenomenon.
>
> To shed light on this point, in an experiment we performed after the submission deadline, we analyzed LIQUID [1]. LIQUID is a unified multimodal-output model that differs from Chameleon only because its LLM backbone was fine-tuned from Gemma-7B, rather than being trained from scratch. In particular, LIQUID encodes the images with the same VQ-VAE as Chameleon. In the table below, we report the ablation analysis for the VQAv2 task.
>
> | Liquid | VQA (2000pt) |
> |:-------:|:------------:|
> |Baseline| 0.576 |
> |EOI | 0.543 |
> |Last | 0.549 |
> |img->text | **0.177**|
>
> This experiment shows that having a VQ-VAE encoder is not sufficient to produce the narrow gate, suggesting that the natively multimodal training process and objective function play crucial roles.
>
> 2. > *[...] the contributions are somewhat difficult to assess. The narrow gate token [...] is a widely observed phenomenon in both LLMs and VLMs.*
>
> We discuss the memory tokens in lines 912-921 and briefly cite the analogy with the narrow gate in lines 331-333.
> While both the narrow gate and memory tokens are known to store global input information, our study goes further. Specifically, we show that the [EOI] token in Chameleon and Emu not only stores global information but also emerges as a **unique** cross-modal communication channel in these multimodal models. Previous studies focusing on registers [2] only addressed VLMs trained for image understanding, where **several register tokens** store global information. The phenomenon we describe in this work is novel both in the model we analyze and in the qualitative picture that emerges on such multimodal-output VLMs.
> Secondly, unlike memory tokens that are often manually introduced to capture global information, either through explicit design [3] or training objectives [4], the [EOI] token becomes a *“narrow"* gate for the information flow naturally during the autoregressive training, without targeted intervention. To our knowledge, this emergent role of [EOI] has not been found in existing work in either unimodal or multimodal output VLMs. If the reviewer can suggest relevant literature, we will be happy to compare and discuss it further in the camera-ready version of the work.
>
> > *After identifying the narrow gate token pattern, I believe the paper should go further by proposing ideas to mitigate the drawbacks of this pattern and support them with experimental validation—even small-scale experiments would be valuable. [...]*
>
> Following the reviewer’s suggestion, we experimentally validated our proposed mitigation strategy by fine-tuning both Chameleon-7b and Emu3. We trained the models on a dataset composed of 10k samples from the LLaVA training set and 5k samples from the VQAv2 training set. We fine-tuned the model for one epoch using LoRA with a batch size of 128, weight decay of 0.1, and a cosine-annealed learning rate starting at 2e-5. During training, we masked the attention to the [EOI] token.
>
> To evaluate performance, we used 50 validation samples each from VQAv2 and COCO. Results are summarized below:
> - **Chameleon**:
>     - VQAv2 accuracy improved from 50% to 62% (base). With ablation of [EOI], it improves from 0.05% to 52%.
>     - COCO-CIDER score increased from 0.48 to 0.72 (base). With ablation of [EOI], the score increased from 0.06 to 0.66.
>
> - **Emu3**:
>     - VQAv2 accuracy improved from 44% to 50% (base). With ablation of [EOI], it improves from 35% to 46%.
>     - COCO-CIDER score increased from 0.73 to 0.87 (base). With ablation of [EOI], the score increased from 0.40 to 0.82.
>
> To validate the importance of keeping [EOI] masked during training, we performed a standard fine-tuning run without masking [EOI]. Then, we repeat the ablation experiment in Tab. I and we measured the effect of ablation [EOI] after the fine-tuning.
> - **Chameleon**: Final VQAv2 accuracy was 53% (base) and 35% with [EOI] ablation. COCO-CIDER was 0.75 (base) and 0.05 with [EOI] ablation.
>
> - **Emu3**: Final VQAv2 accuracy was 45% (base) and 43% with [EOI] ablation and COCO-CIDER was 0.82 (base) and 0.48  with [EOI] ablation.
>
> These results show that the training with [EOI] masked is an effective strategy to remove almost completely the gating mechanism of [EOI]. Notably, it can also improve performance versus standard finetuning except for COCO-CIDER in Chameleon.
>
> 3. > *The experiments in the paper are limited to the image classification task [...]*
>
> The main findings of the paper are summarized in Table I of Section. 3.3 and support the central role of the narrow-gate ([EOI]-token) in the Chameleon and Emu models across diverse tasks: ImageNet classification, visual question answering (VQAv2), and caption generation (Flicker-30k and MS-COCO). Therefore, our experiments are not limited to image classification tasks.
>
> To extend the experimental evidence given for ImageNet in Sec. 3.2, we analyzed the attention patterns of Fig. 3 for visual question answering (VQAv2) and captioning (Flickr-30k), finding the same characteristic attention pattern.
> The experiment depicted in Figure 4 requires instead the selection of a set of representative concepts associated with a dataset. This selection was simple for Imagenet, where the classes give the meaningful semantics, but it is less trivial in Flickr30k (or VQAv2).
>
> To address the reviewer's concern, we extend the experiments to a subset of 10000 samples taken from the Flickr30k dataset, analyzing the Chameleon-7b representations.
> To extract the essential concepts contained in the captions, we used a gte-Qwen1.5-7B-instruct [5] (a Sentence Transformer [6]) to map each caption to a vector encoding its semantics. We then measure the similarity between the hidden representations of [EOI] and the Flickr30k embeddings from the Sentence Transformer with the Neighborhood Overlap [7]. If an internal representation of Chameleon is similar to the embeddings from the gte-Qwen1.5-7B-instruct this means that these representations encode similar abstractions of the image.
> The result of this experiment is consistent with the one on ImageNet: the similarity between [EOI] and the captions embedding increases from 0.0 to 0.16 between layers 5 and 9, and it remains approximately 0.16 for all the remaining layers. For the average image embedding, and all the other image embeddings, it remains close to zero.
>
> In addition, to further validate the generality of our results within the classification domain itself, we have introduced two additional benchmarks: Caltech101 [8] and Describable Textures Dataset [9]. Also these cases exhibit trends consistent with those shown in the paper in Fig. 3 for ImageNet.
>
> 4. > *The paper focuses exclusively on the information flow from vision to text. [...]*
>
> This is a relevant and interesting question. However, exploring the narrow gate in image generation tasks involves significant additional experimentation and analysis. It would be best addressed in a separate, future study.
>
>
> [1] Wu, Junfeng, et al. "Liquid: Language models are scalable and unified multi-modal generators."
>
> [2] Neo et al. Towards Interpreting Visual Information Processing in Vision-Language Models.
>
> [3] Wen et al. Efficient vision-language models by summarizing visual tokens into compact registers.
>
> [4] Burtsev et al. Memory transformer.
>
> [5] huggingface.co/Alibaba-NLP/gte-Qwen1.5-7B-instruct
>
> [6] Reimers et al., Sentence-BERT: Sentence Embeddings using Siamese BERT-Networks
>
> [7] Doimo et al., Hierarchical nucleation in deep neural networks
>
> [8] Li et al, Caltech 101,  doi.org/10.22002/D1.20086
>
> [9] M. Cimpoi, et al. Describing Textures in the Wild
>
> ---
> Thank you again for your detailed suggestions. If we manage to address your main concerns, we politely ask you to consider raising your score. If, on the other hand, there are some remaining questions, we are happy to address them during this rebuttal discussion period.

---

> > ### Comment · Reviewer_R1ay · 2025-08-03
> >
> > Thank you for your response. The additional explanations and experimental results have successfully addressed my concerns regarding the novelty, application value, and scenario richness of this work. "The narrow gate" is an interesting phenomenon in image-text information transmission, and you have thoroughly verified it across multiple tasks, benchmarks, and baselines. Therefore, I have decided to raise my rating to 4. I hope the authors can revise the paper in accordance with the responses in the rebuttal section.
> >
> > Additionally, I have a new question: Does the newly added experiment in the rebuttal section, which involves "masking the attention to the [EOI] token during training," suggest that multimodal large models may NOT need to add the two special tokens  [SOI] [EOI] before and after the input image information when formating the multi-modal inputs? If the [EOI] token is removed, will the text's attention to the image part be distributed across image tokens, or will it still focus on the token after the image (such as possibly a "\n" newline character instead of the removed [EOI])?

---

> > > ### Author Response · Authors · 2025-08-04
> > >
> > > Thank you for recognizing the contributions and novelty of our work. We appreciate your decision to raise your score to 4, and will add the fine-tuning experiment and related insights from the rebuttal into the revised manuscript.
> > >
> > > Below, we address your additional question.
> > >
> > > We believe that, in a multimodal generation setting, a separator between modalities is useful even if it does not act as a collector of image semantics. It helps the model to understand when the image generation part is over and the text generation begins, and vice versa, in a way that is analogous to the user/assistant special tokens in the chat templates.
> > >
> > >  > *Does the newly added experiment suggest that multimodal large models may NOT need to add the two special tokens [SOI] [EOI]*
> > >
> > > To address your first question, we removed [EOI]  from the input prompt in the original Chameleon-7B and observed a performance drop in COCO-CIDER from 0.48 to 0.02 (measured in 300 examples). This is, incidentally, an experiment very similar to the zero ablation experiment reported in Table 1.
> > > In the Chamelon-7B finetuned for this rebuttal, removing entirely the [EOI] tokens leads to only a 20% performance drop in COCO CIDER. This indicates that the fine-tuned model may no longer need the [EOI] token to perform relatively well.
> > >
> > > > *If the [EOI] token is removed, will the text's attention to the image part be distributed across image tokens*
> > >
> > > We repeated the experiment from Fig. 3 on 300 COCO samples and found that the “\n” token at the end of the image receives the same attention as the full image. As we said in Sec. 3.2 (lines 165-169), this does not imply that “\n” is a narrow information channel, between image and text, like [EOI].
> > > To validate this, we conducted the ablation study presented in Table 1 on “\n”. Removing “\n” does not affect the performance, confirming that it does not introduce the gating effect of [EOI]. We can therefore conclude that the attention from text to the image is more distributed among image tokens.
> > >
> > > Thank you again for the time you spent on this revision and for your appreciation of our work.

---

### Official Review · Reviewer_9UB8 · 2025-06-30

**Clarity:** 3
**Significance:** 2
**Originality:** 3
**Rating:** 3
**Confidence:** 4

**Summary:**

This is a study on the different representation patterns emerging in multimodal LLM (image + text) depending on how they are trained and their architecture. The findings from the authors draw a clear distinction between models trained to generate multimodal outputs compared to models trained to generate unimodal (text-only) outputs. The former maintain a more clear separation per modality in their token latent space and have special tokens, like $[EOI]$, that exhibit a significantly higher amount of attention from text tokens compared to the rest of the multimodal tokens. This does not hold for unimodal out models where instead tokens of different modality become more similar in the deep layer of the model and attention tends to be more scattered around all image tokens. Leveraging this observation the author shows how it is possible to modify the attention pattern towards the [EOI] token in a few MLLMs to destroy their performance.

**Questions:**

* How would Fig. 2 look like for [EOI]? I.e., is the [EOI] token more aligned to text representations wrt to the average image token?

* Have you analized the attention pattern of [EOI] wrt the rest of the image tokens for the different models? Is [EOI] in chameleon somehow attending tot he whoel image, while in the other models is not?

**Ethical Concerns:**

["NO or VERY MINOR ethics concerns only"]

**Final Justification:**

The author clarified a lot of things that were not clear in the main paper and answered most of my doubts, hence I'm raising my score. The extent of additional experiments provided in the rebuttal is great but it's what could have made the submission truly interesting if executed with more time. If this was a journal my rating would have been a major revision, considering it's a conference and that's not possible I would consider this work really borderline.
I don't have major issues with the work being accepted if all the additional results are integrated in a cohesive way, but at the same time the original submission was in my opinion very weak and it would be interesting to re-evaluate a version of the work with the additional material included in a cohesive story. The finetuning experiments in the rebuttal are in my opinion very counter-intuitive and go against the intuition in the paper. I didn't find the explenation of the authors very convincing but to their credits they only had the extremelly limited time of the rebuttal phase to run experiments and interpret the results.

Considering all the points above I would lean towards keeping a borderline negative rating (3).

**Limitations:**

yes

**Paper Formatting Concerns:**

Few suggestions (msotly subjective):

* I think the paper would be more entertaining to read if you mix  explanation of the experiment and metric used with the actual experimental results, instead of having them in 2 separate sections (2-3)

* Fig. 3 and Fig. 4 change the legend from “last image” to “last image token”, same for "first image"

* What is $y$ in Eq. 2? It is not clear from the text at that point in the paper, make it more clear by providing an example or move the definition of Eq. 2 after you define how the experiment is done. At that point it would be clear that $y$ is an imagenet class.

**Quality:**

2

**Strengths And Weaknesses:**

## Strengths

+ The finding that natively multimodal models keep almost orthogonal representation in their token space across modalities is interesting and somehow counterintuitive to me. I would have expected the natively multimodal training to actually help the two modality to converge into some form of shared representation. This result hints at the possibility of having better ways of mixing the two modalities when training natively multimodal out models.

+ The experimental analysis is well structured and nicely self-contained, the work provides all the ingredients to understand its content.

## Weaknesses

a. **Ambiguity in how the cross-modal attention is computed**: I would like to challenge the way cross-modal attention is computed in Eq. 1. This is the main experimental tool used throughout the paper but I would argue that it does not consider few aspects:

1. In my opinion it would make more sense to check the cross-attention pattern only among the predicted tokens belonging to the “answer” to a certain task (e.g., the class in the imagenet examples) rather than the whole text tokens. A lot of text tokens might belong to the task instructions and should naturally not interact much with visual tokens.

2. Influence of text->image attention vs text->text attention when predicting the next token. All the analysis is performed only considering the “text->image” attention and attention scores are also reported normalizing only across cross-modal attention. It might be the case that in absolute terms the cross-modal attention is lower than the uni-modal attention between the current text token being predicted and the previous ones. This would be important to report to understand when the visual content really plays a role and when instead it does not.

3. Attention experiments in Fig. 3 are for a specific “global” task, i.e., image classification. It would be interesting to verify whether exactly the same attention pattern emerges for other visual understanding tasks. Tab. 1 provides some evidence in that regard, but only partial. Same holds for the clustering experiments in Fig. 4 that are fundamental to identify which tokens matter and which do not. Do these hold for tasks different from imagenet?

b. **Are attention patterns really predictive of which tokens act as narrow gates?** The answer the paper implies is yes, but to be honest the evidence from the paper is somehow not convincing.  For example Fig. 3 Chameleon-7B has disproportionally high attention over the “last image” token, significantly higher than the attention to [EOI] token for 2/3 of the model. However the results in Tab. A2 shows that masking the “last image token” has a negative effect way lower than masking [EOI]. Same holds for the 32nd token. This in my opinion points out to the fact that the attention analysis is not enough to identify these narrow gates into the model and something on top is missing. The neighborhood analysis in Fig. 4 might help there, but as mentioned above is only limited to a specific task and it’s not clear how it could be generalized to several.

c. **Actionability of the findings** Building on weakness B, it is not clear to me how the findings of this paper can help us to design better or more robust models going forward. This discussion is currently missing from the paper, reducing the contribution to a detailed experimental report. This alone would not be a major issue, but combined with the doubts on the experimental procedure I raised in the previous two weaknesses, it makes me question the completeness of the work.

d. [minor]**Influence of the visual encoder vs the training loss**: the authors did touch upon this discussion in the paper, but only partially. A side effect of the models chosen for this evaluation is that models trained to be unimodal output have a strong vision encoder trained already for image understanding (Siglip or Pixtral-ViT), while models trained for multimodal generation have typically VQVAE which have never been trained for being powerful standalone image feature extractor. The results on Janus seem to suggest that this is the main differentiation, rather than the training objective of the MLLM. A model like [1] could be included in the discussion to either support or deny this claim as similar to Janus uses a SigLIP encoder and has been trained for both text and image output, but without having a separate visual encoder used for generation.

## References

1. [Tong, Shengbang, et al. "Metamorph: Multimodal understanding and generation via instruction tuning." arXiv preprint arXiv:2412.14164 (2024). ](https://arxiv.org/pdf/2412.14164)

---

> ### Author Rebuttal · Authors · 2025-07-31
>
> We thank the reviewer for their comments that our results and analysis are both well structured and surprising, and that they could help design better ways to train natively multimodal models. In the section below, we will address your concerns.
>
> ###  Weaknesses
>
> > *a.  Ambiguity cross-modal attention is computed:*
>
> - 1: The cross-attention pattern reported in Fig. 3 remains quantitatively consistent if we analyze only the answer tokens as the reviewer suggests. To answer the reviewer's concern, we repeated the experiment described in Sec. 3.2 on Chameleon-7b, measuring the attention patterns only on the token generated after the prompt “ `<image>` This animal is a _”.
> Also in this setup, the highly attended image tokens are the 32nd, [EOI], and last image ones, with [EOI] receiving 44% on the total cross modal attention between layers 1 and 8, the 32nd receiving 54% of the cross modal attention between layer 5 and 12 and the last token 73 % of the total cross modal attention after layer 12.
> Therefore, the highly attended tokens do not change if we restrict the analysis to the answer tokens.
>
> - 2: In Llava and Chameleon-7b, the unimodal “text->text” attention makes up from 12% to 24% of the total attention throughout the model, much smaller than the cross-modal part. The general patterns of the cross-modal part remain unchanged when we include the unimodal one.
> We’d like to highlight, however, that the primary focus of our work is on the communication of information between image and text in Vision-Language Models. The answers can be grounded on the visual part of the input only through the cross-modal part of the attention, regardless of how strong it is relative to the unimodal attention. For this reason, in Fig.3, we report where the information flows in the cross-modal attention block.
>
> - 3: We performed additional experiments to verify that the patterns of Fig. 3 are task agnostic and remain qualitatively unchanged in the captioning COCO, Flickr, and visual understanding VQAv2 tasks.
> The experiment depicted in Figure 4 requires the selection of a set of representative concepts associated with a dataset. This selection is simple for Imagenet, where the meaningful semantics is given by the classes. It is less trivial in Flickr30k (or VQAv2). To address the reviewer's concern, we extend the experiments to a subset of 10000 samples taken from the Flickr30k dataset. To extract the important concepts contained in the captions, we used a gte-Qwen1.5-7B-instruct[1] (a Sentence Transformer [2]) to map each caption to a vector encoding its semantics. We then measure the similarity between the hidden representations of [EOI] and the Flickr30k embeddings from the Sentence Transformer. We compute the representation similarity with the Neighborhood Overlap. The result of this experiment is consistent with the one on ImageNet: the similarity between [EOI] and the captions embedding increases from 0.0 to 0.16 between layers 5 to 9, and it remains approximately 0.16 for all the remaining layers. For the average image embedding, and all the other image embeddings, it remains close to zero. \
> We also extended the analysis to two other classification tasks: Caltech101[3], comprising 101 classes of different objects, and DTD Dataset [4] with 47 classes of textures inspired by human perception. Also in these cases, the overlap profiles exhibit trends consistent with COCO captions and ImageNet.
>
> > *b. Are attention patterns predictive*
>
> As we write in lines (165, 166),  a candidate's narrow gates must *"(i) have a large weight in the text-image attention, and ii) have a rich semantic knowledge of the image"*. Therefore, **we never implied that attention patterns alone are predictive of tokens acting as narrow gates. The analysis of Sec. 3.2 is aimed at discussing both these two necessary conditions**, with the section paragraph *“Probing the semantic content of visual tokens”* showing that in Emu3 and Chameleon, only [EOI] retains visual semantic information and satisfies condition (ii) and can thus be considered a candidate narrow gate. The final evidence that [EOI] acts as a narrow gate is given by the ablation results shown in Table 1. By addressing your previous concern a) we also showed that the analysis of Sec. 3.2 generalizes across datasets (Caltech101, DTD) and tasks (COCO-captioning).
>
> > *c. Actionability of the findings*
>
> Between lines 323 and 333, we discuss several potential practical implications of our findings, suggesting that a possible way to make the models more robust and *“encourage a more distributed communication can be masking the text-on-[EOI] attention during the last part of training”* (lines 325-326).
> To validate experimentally this claim and address the reviewer's concern, we fine-tuned Chameleon-7b and Emu3 on a dataset composed of 10k samples from the LLaVA training set and 5k samples from the VQAv2 training set, masking the attention to [EOI] during training.
> To evaluate performance, we used 50 validation samples each from VQAv2 and COCO. Results are summarized below:
> - **Chameleon**:
>     - VQAv2 accuracy improved from 50% to 62% (base). With ablation of [EOI], it improves from 0.05% to 52%.
>     - COCO-CIDER score increased from 0.48 to 0.72 (base). With ablation of [EOI], the score increased from 0.06 to 0.66.
>
> - **Emu3**:
>     - VQAv2 accuracy improved from 44% to 50% (base). With ablation of [EOI], it improves from 35% to 46%.
>     - COCO-CIDER score increased from 0.73 to 0.87 (base). With ablation of [EOI], the score increased from 0.40 to 0.82.
>
> To validate the importance of keeping [EOI] masked during training, we performed a standard fine-tuning run without masking [EOI]. Then, we repeat the ablation experiment in Tab. I and we measured the effect of ablation [EOI] after the fine-tuning.
> - **Chameleon**: Final VQAv2 accuracy was 53% (base) and 35% with [EOI] ablation. COCO-CIDER was 0.75 (base) and 0.05 with [EOI] ablation.
>
> - **Emu3**: Final VQAv2 accuracy was 45% (base) and 43% with [EOI] ablation and COCO-CIDER was 0.82 (base) and 0.48  with [EOI] ablation.
>
> These results show that the training with [EOI] masked is an effective strategy to remove almost completely the gating mechanism of [EOI]. Notably, it can also improve performance versus standard fine-tuning except for COCO-CIDER in Chameleon. The masked-finetuning, targeted on [EOI], is an example of how the findings of this work can be used to improve the model's performance and robustness.
>
> > *d.  Influence of the visual encoder*
>
> We agree with the reviewer's observation that the models analyzed can not fully disentangle the impact of the vision encoder from that of the training protocol. To shed light on this point, in an experiment we performed after the submission deadline, we analyzed LIQUID [5]. This model is unified multimodal in output, differing from Chameleon only because its LLM backbone **was finetuned from Gemma-7B** and rather than being trained from scratch. In particular, **LIQUID encodes the images with the same VQ-VAE as Chameleon**. In the table below, we report the ablation analysis for the VQAv2 task. The results indicate that LIQUID behaves similarly to Janus and the other unimodal-output models: removing the [EOI] token does not affect performance, whereas removing the internal image tokens does degrade it. This experiment shows that having a VQ-VAE encoder is not sufficient to produce the narrow gate, suggesting that the natively multimodal training process and objective function play crucial roles.
>
> | *Liquid*      | *VQA (2000 pt)* |
> |---------------|-----------------|
> | *Baseline*    | 0.576           |
> | *EOI*         | 0.543           |
> | *Last*        | 0.549           |
> | *img->text*   | **0.177**           |
>
>
> ### Questions
>
> > *How would Fig. 2 look like for [EOI]?...*
>
> We have measured the median cosine similarity between the [EOI] embeddings and i) random text token embeddings and ii) random image token embeddings chosen from 200 random samples of the Flickr dataset. For the Chameleon 7b model, both profiles start similarly, with an initial peak at 0.10 at layer 2. After layer 10, the cosine similarity with image embeddings drops to 0.01, and the cosine similarity with text remains constant at 0.10. Both these values are well below the average text-text and image-image similarity.
>
> > *Have you analyzed the attention pattern of [EOI] wrt the rest of the image tokens for the different models?...*
>
> This seems not to be the case. To evaluate the sparsity of the attention pattern of the [EOI] tokens into the image, we calculated the (normalized) entropy of the [EOI] row of the attention pattern. We have done so for the Chameleon 7b and Llava models.
> In the former - even if we exclude the anomalous contribution from the 32nd token  - Snorm fluctuates between 0.5 and 0.82.
> However, Snorm is much higher in the Llava model, with values fluctuating between 0.73 and 0.99, indicating that in some layers the attention from [EOI] in Llava is almost uniform.
>
> ### Paper formatting concerns
> > *Fig. 3 and Fig. 4 change the legend...*
>
> Thank you, we will update the legend in the camera-ready version.
>
> > *What is  in Eq. 2?...*
>
> Thank you for raising this clarity issue. We will add an example in the method section of the paper after Eq.2 is defined.
>
> [1] huggingface.co/Alibaba-NLP/gte-Qwen1.5-7B-instruct
>
> [2] Reimers et al., Sentence-BERT: Sentence Embeddings using Siamese BERT-Networks
>
> [3] Li et al, Caltech 101, doi.org/10.22002/D1.20086
>
> [4] M. Cimpoi, et al. Describing Textures in the Wild
>
> [5] Junfeng, et al. "Liquid: Language models are scalable and unified multi-modal generators."
>
> ---
>
> Thank you again for your detailed suggestions. If we manage to address your main concerns, we politely ask you to consider raising your score. If, on the other hand, there are some remaining questions, we are happy to address them during this rebuttal discussion period.

---

> > ### Author Response · Authors · 2025-08-05
> > **Response Summary + Remaining Questions**
> >
> > As the discussion period is coming to an end, we kindly ask you whether your concerns have been addressed by our responses. \
> > We summarize them below.
> >
> > a. **Concerns about cross-modal attention**:
> >
> > We show that:
> > 1. The attention remains consistent if we focus only on the answer tokens;
> >
> > 2. The text-->text attention is much smaller than the cross-modal ones;
> >
> > 3. The attention patterns of Fig.3 and the semantic analysis of Fig.4 are consistent across other datasets (Flickr30, COCO, VQAv2), tasks (captioning), and classification benchmarks (Caltech101, DTD);
> >
> > b. **Attention and narrow gates**: We clarified that high attention is not a sufficient condition for a narrow gate;
> >
> > c. **Actionability**: We reported a small-scale fine-tuning experiment based on the narrow gate, showing how our findings can improve the robustness of multimodal models;
> >
> > d. **Visual encoder vs training loss**: We used LIQUID to show that training from scratch is likely the main factor behind the narrow gate formation;
> >
> > We thank you again for the time spent reviewing our paper. Please let us know if anything remains unclear.

---

> > > ### Comment · Reviewer_9UB8 · 2025-08-05
> > > **Reply to auhors rebuttal**
> > >
> > > Thank you for the detailed response and sorry for the delay, discussion period is however being extended.
> > > Few follows up
> > >
> > > a. Thanks for the additional experiments supporting the initial observations, I would suggest to integrate them in a revised version fo this submission.
> > >
> > > b. ack, thanks for the clarification. Verifying (ii) it's harder for tasks different from classification as explained by the authors in reply to (a.-3), however if the experiments in reply (a.) show that the findings on classifcation tasks somehow generalize also to other tasks this should not be an issue.
> > >
> > > c. why would masking [EOI] attention during SFT result in increased performance? Accoridng to the paper interpretation [EOI] is the crucial gate of communicaiton among modalities. not giving access to it should make SFT harder? Any intuition? This experimental observation is a bit un-intuitive given the story of the paper.
> > >  It also seems that even without the masking of [EOI] fine tuning does ameliorate the drop in performance when masking [EOI] at inference time. Any intuition on why this would be the case?
> > > Generally speaking these results and analysis would have made the original submission way more interesting so I woudl suggest to add them to a revision with a proper comment.
> > >
> > > d. I'm not sure this addresses my concern. If LIQUID is equivalent to chameleon wrt the visual encoder it does not show wheter the distribution of the attention comes from the pre-trainign of the vision encoder or from the training objective?
> > >
> > > Q1 + Q2 --> Thanks for checking!

---

> ### Author Response · Authors · 2025-08-06
>
> > *a-b. I would suggest to integrate them in a revised version of this submission*
>
> We are pleased to hear that our replies addressed your concerns in a-b. We will add the plots of the experiments in the appendix and discuss them in Sec. 3.2
>
> **Clarification of fine-tuning scope and related questions of point c.**
>
> **Scope.** As a general comment, we would like to emphasize that our fine-tuning experiment was done primarily to show that we can effectively remove the narrow gate, making the cross-modal communication more distributed. The performance increase was not the main goal, only a side remark we made.
>
> **Questions:**
>
> > *c. why would masking [EOI] attention during SFT result in increased performance? [...] Any intuition?*
>
> A possible intuitive explanation is as follows:\
> When EOI is masked, the model is forced to learn a new way to communicate the visual information without relying on EOI. During inference, EOI is unmasked and will also communicate its own visual information, useful for the downstream task. If the model can combine redundant useful information to solve a task, this generally helps generalization [1, 2]. This can be intuitively seen as a form of (implicit) ensembling of predictors that is known to improve the performance of ML models. \
> However, this is only a speculation, and more evidence and analysis are needed to validate it.
>
>
>
> >  *c. not giving access to it should make SFT harder?*
>
> If by harder we mean that the training loss is higher at the beginning of training and the convergence is slower, then yes, the masked training takes more iterations to converge. However, even though masking [EOI] may slightly delay convergence, the model can still learn to route information effectively through different pathways.
>
> > *c. It also seems that even without the masking of [EOI], fine-tuning ameliorates the drop in performance*
>
> This is not the case for all tasks.
>  In COCO captioning, fine-tuning without masking does not reduce the drop in performance:
>
> - Chameleon: performance gap increases from 0.48 → 0.06 (before) to 0.75 → 0.05 (after)
>
>  - Emu3: remains similar, from 0.73 → 0.40 (before) to 0.82 → 0.48 (after)
>
> For VQAv2, we fine-tuned on some examples from that dataset. This likely helped the model learn task-specific conventions (e.g., short, one-word answers), which likely made the validation drop smaller even without masking EOI.
>
> > *Generally speaking these results and analysis would have made the original submission way more interesting so I woudl suggest to add them to a revision with a proper comment.*
>
> We will add the fine-tuning experiments in a dedicated section (3.4) at the end of the results.
>
>
> > *d. I'm not sure this addresses my concern. If LIQUID is equivalent to chameleon wrt the visual encoder, it does not show whether the distribution of the attention comes from the pre-training of the vision encoder or the training objective?*
>
> In our previous answer, we tackled the question of whether the main discriminating factor for the narrow gate is the training objective or a powerful (high-level) image feature extractor. We used LIQUID because we believe it can better disentangle three factors:
>
>  1. training objective (multimodal output vs text-only output);
>
>  2. training “history” (backbone trained from scratch vs fine-tuned on pre-existing LLM);
>
>  3. vision encoder (low-level tokens vs text-aligned “semantic rich” tokens);
>
> LIQUID, Metamorph, Chameleon, and Emu all share the same training objective in that they can generate multimodal outputs within a unified architecture.
>
> **LIQUID differs** from Chameleon/Emu only in its **training “history”**: LIQUID’s backbone is fine-tuned from an existing LLM, while Chameleon/Emu are trained from scratch. Our previous reply shows that this is sufficient to remove the modality gap and the narrow gate in this model.
>
> **Metamorph** has *two* elements that differ from Chameleon/Emu: **training history and vision encoder**. First, its backbone is fine-tuned from an existing LLM (as is the case for LIQUID). Second, it uses a SigLIP encoder, which produces more “text-aligned” embeddings due to the contrastive pretraining.
> Given that the first condition is sufficient to suppress the narrow gate, and further, all the models trained with strong vision encoders we analyzed (LLaVA, Pixtral, Janus) had distributed image-text communication patterns, we do not expect that Metamorph will have a narrow gate token.
>
> Please let us know if we addressed your concerns.
>
> [1] Doimo et al., *Redundant representations help generalization in deep neural networks*, NeurIPS 2022
>
> [2] N Srivastava et al., *Dropout: A Simple Way to Prevent Neural Networks from Overfitting*, JMLR 2014

---

> > ### Comment · Reviewer_9UB8 · 2025-08-07
> > **Reply**
> >
> > > However, this is only a speculation, and more evidence and analysis are needed to validate it.
> >
> > Agreed. This type of experiments would have made a way more comprehensive submission with more actionable outcomes for practitioners.
> >
> > > If by harder we mean that the training loss is higher at the beginning of training and the convergence is slower, then yes, the masked training takes more iterations to converge.
> >
> > Does this mean that the two reported results above are trained for a different number of iterations?
> >
> > > LIQUID
> >
> > Thanks for the additional explanation. I now understood the motivation for the experiments.

---

> ### Author Response · Authors · 2025-08-07
>
> > *Does this mean that the two reported results above are trained for a different number of iterations?*
>
> No, both training runs were done on the same 15k examples for one epoch for a total of 117 gradient updates. \
> Inspecting the training loss of intermediate checkpoints, we noted that when the training is masked, more gradient updates were needed to converge.
>
> > *This type of experiments would have made a way more comprehensive submission*
>
> Thank you for appreciating the value of this experiment. We are happy to include it in the camera-ready version as an interesting application of our main finding about the narrow gate in VLMs.
>
> We are also grateful for the time you spent on this review and your comments. Both helped us improve the quality of this work.

---

### Official Review · Reviewer_7J47 · 2025-07-01

**Clarity:** 3
**Significance:** 3
**Originality:** 2
**Rating:** 5
**Confidence:** 4

**Summary:**

This paper investigates how multimodal-output VLMs that both generate images and text differ from text-only output VLMs in the way they process and transfer visual information into the textual domain. It shows that multimodal-output VLMs (e.g., Chameleon, Emu3) maintain largely separate image and text embeddings and rely on a single [EOI] token as a *narrow gate* through which all visual semantics must pass. By analyzing cross-modal attention (Sec. 3.2), the authors demonstrate that [EOI] token captures a disproportionately large share of text-on-image attention and retains the bulk of semantic content beyond early layers. Ablation (Sec. 3.3) and steering (Sec. 3.4) experiments confirm the causal importance of the [EOI] gate—blocking its connections drastically degrades performance on VQA, captioning, and classification tasks, whereas ablating all other visual tokens has a smaller effect.

**Questions:**

- At lines 288–290, you claim that visual information is passed through the last token. To substantiate this, one must examine the value of the last token’s \(\mathcal{X}^{l,\text{text\_gt}}\). Have you checked this? Do the results actually support your claim?
- The notation \(A_{\text{img}\rightarrow\text{text}}\) on line 88 is ambiguous. Is it meant to represent a set, a matrix, or something else? How is this symbol defined and used in the main text? It appears not to be referenced elsewhere in the paper.
- In the Chameleon model, what is your hypothesis for why the 32nd token serves as the narrow gate? What underlying mechanism or training dynamic leads this specific position to assume gating behavior?

**Ethical Concerns:**

["NO or VERY MINOR ethics concerns only"]

**Final Justification:**

This paper investigates how multimodal-output VLMs and text-output VLMs exchange information across modalities. By identifying the narrow gate token, the authors demonstrate how visual semantics propagate differently depending on the model type (i.e., multi-modal output VLMs versus text-only output VLMs).

My initial major concern was the limited novelty of the work, as the original submission focused primarily on establishing the existence of the narrow gate without offering deeper or unexpected insights.

While I still believe that the paper does not fully explain the underlying mechanism behind the narrow gate phenomenon, my main concern regarding limited novelty has been sufficiently addressed. The additional experiments presented in the revised version highlight the importance of the narrow gate and point to meaningful directions for future research. Accordingly, I find a final score of 5 to be appropriate.

**Limitations:**

Yes

**Quality:**

4

**Strengths And Weaknesses:**

## Strengths
To the best of my knowledge, this paper is the first to compare how multimodal-output VLMs and text-only VLMs exchange information across modalities. By identifying a single *narrow gate* token (e.g., [EOI]) through which visual semantics flow into the text pipeline, it lays the groundwork for future interpretability studies of unified vision–language models. As such, it offers a novel framework that researchers can build on to better understand and diagnose multimodal architectures.

## Weaknesses
- Limited scope of findings: the paper compares multimodal‐output and text‐only VLMs only at the token level, centering its analysis on the “narrow gate” token. It focuses almost entirely on demonstrating the gate’s existence and lacks any further substantive or surprising conclusions.
- Questionable validity of the zero‐ablation method: the zero‐ablation approach may distort the model’s representation distribution, undermining the reliability of the results presented in Table 1.
- No causal analysis of narrow‐gate emergence: although the study highlights the narrow gate and argues that training differences give rise to it, it does not rigorously investigate or empirically validate the mechanisms by which this bottleneck forms, leaving its origin unexplained.

---

> ### Author Rebuttal · Authors · 2025-07-31
>
> We thank the reviewer for their recognition of the originality of our findings and the possible future applications for a better understanding of MMLMs. Below, we address your concerns.
>
> ### Weaknesses
> > *Limited scope of findings: the paper [...] focuses almost entirely on demonstrating the gate’s existence and lacks any further substantive or surprising conclusions.*
>
> We believe that our paper goes beyond demonstrating the existence of the *“narrow gate”* at the [EOI] token.
> First of all, this finding is tightly related to the general, and counterintuitive, observation that native multimodal-output models maintain a large modality gap across all layers (Fig. 3), a property generally considered to be a drawback for cross-modal reasoning [1,2].
>
> We support our findings by analyzing the impact of diverse architectures and training protocols on the narrow gate phenomenon. These insights are of practical importance. Localizing and characterizing how these models develop an effective communication mechanism despite the modality gap is a finding central and timely in the multimodality community,  since this class of models is being employed in several state-of-the-art multimodal training pipelines [3,4], motivated by advances in autoregressive image generation [5].
>
> We discuss the potential applications where the narrow gate can be exploited proactively in Sec. 3.4 (e.g., semantic steering via activation patching) and propose how to use it to reduce memory and computing usage. We also suggest that a possible way to make the models more robust and *“encourage a more distributed communication can be masking the text-on-[EOI] attention during the last part of training”* (lines 325-326).
>
> To validate experimentally the last claim and address the reviewer's concern, we fine-tuned Chameleon and Emu3, masking the [EOI] tokens on an instruction dataset composed of 10k samples from the LLaVA training set and 5k samples from the VQAv2 training set. In this case, we can significantly reduce the detrimental effect of [EOI] ablation on performance. Results are summarized below:
> - **Chameleon**:
>   - VQAv2 accuracy improved from 50% to 62% (base). With ablation of [EOI], it improves from 0.05% to 52%.
>   - COCO-CIDER score increased from 0.48 to 0.72 (base). With ablation of [EOI], the score increased from 0.06 to 0.66.
>
> - **Emu3**:
>   - VQAv2 accuracy improved from 44% to 50% (base). With ablation of [EOI], it improves from 35% to 46%.
>   - COCO-CIDER score increased from 0.73 to 0.87 (base). With ablation of [EOI], the score increased from 0.40 to 0.82.
>
> To validate the importance of keeping [EOI] masked during training, we performed a standard fine-tuning run without masking [EOI]. Then, we repeat the ablation experiment in Tab. I and we measured the effect of ablation [EOI] after the fine-tuning.
> - **Chameleon**: Final VQAv2 accuracy was 53% (base) and 35% with [EOI] ablation. COCO-CIDER was 0.75 (base) and 0.05 with [EOI] ablation.
>
> - **Emu3**: Final VQAv2 accuracy was 45% (base) and 43% with [EOI] ablation, and COCO-CIDER was 0.82 (base)  and 0.48 with [EOI] ablation.
>
> These results show that masking [EOI] during training is crucial for removing almost completely the gating mechanism, distributing the communication of visual information across multiple tokens, and removing the issues associated with a constrained information flow (see lines 324-327). Notably, it can also improve performance versus standard fine-tuning, except for COCO-CIDER in Chameleon.
>
> Our results, therefore, provide two main contributions. First, a new empirical finding and a mechanistic explanation for how native multimodal training impacts information flow, advancing the understanding of VLM internals. Secondly, they suggest practical, actionable strategies for exploiting or mitigating the issues associated with the presence of the narrow gate.
>
> > *Questionable validity of the zero‐ablation method [...].*
>
> The zero-ablation approach may distort the model's internal representations. However, for all the models, the zero-ablation on any single token (besides [EOI] in Chameleon and Emu3) does not cause a performance drop despite the distribution shift caused by the ablation (see Tables 1 and A2). This suggests that the performance drop observed in Emu3 and Chameleon when [EOI] is zero-ablated is not due to OOD effects, but instead to the gating mechanism of  [EOI].
>
> We agree with the reviewer that milder ablation techniques exist in the literature, the mean-ablation being one of the most common alternatives [6]. In this case, the ablated token is substituted with its mean value taken over the dataset. We applied the mean ablation to [EOI] in the COCO dataset in Chameleon-7b, and also in this case, we observed a degradation in CIDER from 0.46 to 0.08.
>
> > *No causal analysis of narrow‐gate emergence [...].*
>
> In our study, we analyzed VLMs composed of a different set of architectures (visual encoders, connectors) and trained with different pipelines (from scratch or fine-tuned on existing LLM backbones).
> This wide variety of models allowed us to pin down the conditions to observe a narrow gate: the model must be trained from scratch on multimodal data, both for understanding and generation tasks, and must possess a low-level vision encoder (VQ-VAE). Studying how the narrow gate forms during training is beyond the scope of our work due to the lack of intermediate training checkpoints for Chameleon and Emu3.
>
>
> ### Questions
> > *At lines 288–290, [...] one must examine the value of the last token’s ($\chi^{l,gt}$). Have you checked this?*
>
> We agree with the reviewer on this observation.  We measured $\chi^{l,gt}$   also on the last token, and its value grows from 0 to 0.45 between layer 0 and 24 in Chameleon-7b, from 0 to 0.58 between layer 0 and 44 in Chameleon-34, and from 0 to 0.37 between layer 0 and 11 in Emu3. In all cases, the growth of the  $\chi^{l,gt}$ in the last token comes after the one on [EOI]. When [EOI] is ablated, also the $\chi^{l,gt}$ in the last token drops (see Table 1). Therefore, the claim at lines 288-290 is fully supported by the experiment.
>
> > *The notation ($A_{\text{img}\rightarrow\text{text}}$) on line 88 is ambiguous. [...]*
>
> $A_{i,j}$ is a matrix which we introduce in line 87 as the “attention from token i to token j”. Given that in our setup the image comes before the text (line 85), if we set “$N_{[EOI]} < i ≤ N, 0 < j < N_{[EOI]}$”, $A_{\text{img}\rightarrow\text{text}}$ is the rectangular block of the attention matrix in which the text tokens at positions $N_{[EOI]} < i ≤ N$ attend to the image tokens $0 < j < N_{[EOI]}$. We denote this block of the attention matrix as cross-modal attention (see end of line 86) and analyze it in Sec. 3.2.
> We use $A_{\text{img}\rightarrow\text{text}}$ in Sec. 3.4 at lines 233 - 250 - 255 to reference the part of the attention matrix set to zero when we ablate the communication between image and text tokens.
>
> > *In the Chameleon model, what is your hypothesis for why the 32nd token serves as the narrow gate? [...]*
>
> The 32nd token in chameleon-7b is not a narrow gate. In lines (165, 166), we write that a token must “*(i) have a large weight in the text-image attention, and ii) have a rich semantic knowledge of the image*”. The analysis of Sec. 3.3 shows that condition (ii) is not met for token 32nd, which has a $\chi^{l,gt}$ with the imagenet concepts close to zero in all the layers (see Fig. 4 left).
> In a side experiment, not reported in the paper, we verified that the norm of the token 32 at layers $5-12$ is $\times100$ larger than all the other image tokens. Massively activated tokens in LLMs can occur as artifacts in LLMs, and their origin is the subject of current research [7].
>
> [1] W. Lieng et al., Mind the gap: Understanding the modality gap in multi-modal contrastive representation learning.
>
> [2] Q. Huang et al., Deciphering Cross-Modal Alignment in Large Vision-Language Models with Modality Integration Rate.
>
> [3] https://openai.com/index/introducing-4o-image-generation/
>
> [4] M. Shukor et al., Scaling Laws for Native Multimodal Models
>
> [5] P. Sun et al, Autoregressive Model Beats Diffusion: Llama for Scalable Image Generation
>
> [6] Wang,  et al. "Interpretability in the wild: a circuit for indirect object identification in GPT -2 small."
>
> [7] Sun, M. et al, Massive Activations in Large Language Models
>
> ---
>
> Thank you again for your comments and suggestions. If we manage to address your main concerns, we politely ask you to consider raising your score. Please don’t hesitate to let us know if you have further questions. We are happy to answer your questions during this rebuttal discussion period.

---

> > ### Comment · Reviewer_7J47 · 2025-08-01
> >
> > # W1
> > Thank you for conducting the additional experiments. As I was reading the paper, I also found that empirically validating the hypothesis in Lines 326–327 would provide interesting insight into the narrow gate. I believe the results strongly support the potential role of the narrow gate in this area. Would it be possible to incorporate this experimental finding directly into the main body of the paper, rather than leaving it as part of the discussion only?
> >
> > # W2
> > Thank you for re-running the experiments using mean ablation. Regarding the performance degradation you reported (0.46 → 0.08), could you clarify whether the 0.46 score corresponds to the MS-COCO result of Chameleon-7B without [EOI] ablation? If so, could you explain why this value differs from the 0.48 reported in Table 1 of the paper?
> >
> > # W3
> > I understand that explaining how the narrow gate emerges during training is out of the scope of this study. However, I still believe that the central aim of the paper is to demonstrate the existence of the narrow gate. According to your findings, the *conditions* required for a narrow gate to emerge appear to be well-defined: the model must be trained from scratch on multimodal data for both understanding and generation tasks, and must include a low-level vision encoder (e.g., VQ-VAE).
> >
> > That said, I believe the paper should at least discuss the *mechanism* (not conditions), even speculative, by which these conditions result in the formation of the narrow gate.
> > In the experiment described in W1, you showed that masking the narrow gate can lead to improved performance. If so, why would a VLM learn to rely on the narrow gate in the first place? Without a discussion (or at least a hypothesis) on why the narrow gate emerges, I am left wondering whether it is a phenomenon specific only to a subset of models, rather than a general architectural or training effect.
> >
> > # Q1 & Q3
> > My first and third questions have been resolved thanks to the authors' explanation. I appreciate the detailed clarification.
> >
> > # Q2
> > My second question has also been addressed. However, I would like to point out that the expression $A_{\text{image} \rightarrow \text{text}}=A_{i,j}$ may cause confusion, as it can be misinterpreted to refer to individual scalar values. Since this term plays an important role in the paper, I suggest revising the notation in the final version to make the definition clearer.

---

> ### Author Response · Authors · 2025-08-02
> **Responses**
>
> We are deeply grateful for your timely response and the appreciation of our replies.
>
>
> ### **Response to W1**
>
> Yes, we can and will incorporate this experiment after section 3.4 as section 3.5.
>
> ### **Response to W2**
>
> The value 0.46 reported in this rebuttal was measured on 300 MS–COCO examples. In Table 1, the experiment was run on 2000 examples. This was done to speed up the experiment in the rebuttal phase.
>
> ### **Response to W3**
>
> Our hypothesis for the emergence of the narrow gate stems from the strong modality gap observed in native multimodal models.
>
> This gap is due to the VQ-based image tokenizer, which is known to encode low-level information from the images, while the tokenized words capture higher semantic information  [1, 2]. This different degree of abstraction between text and image tokens is, by construction, present also in the last hidden representations, which are decoded back into pixels and words via the VQ-decoder and the unembedding matrix.\
> Our first finding, shown in Fig. 2, is that this modality gap is also evident throughout all hidden layers in models like Chameleon and Emu3. We observe that image and text embeddings form distinct homogeneous clusters, with low cross-modal cosine similarity, indicating that the modalities follow separate paths occupying different regions in the embedding space.
>
> Given this separation, some mechanism or mediator is needed to communicate the visual information to the text embeddings. This role cannot be played by tokens within the images or text sentences. Due to the autoregressive nature of the transformer, image tokens only generate the following image token, and text tokens do the same for text.
> Since the residual stream maintains the modality-specific separation, and the attention heads themselves must have at least some modality-specific role, the cross-modal interactions are limited.
>
> This leaves the end-of-image special token [EOI] as a unique candidate for cross-modal mediation. Since [EOI] is not aligned with either image or text embeddings, it can serve as a bridge. Supporting this, we find that the cosine similarity between [EOI] and both image and text embeddings is below 0.1, much lower than typical intra-modal similarities.\
> This suggests that [EOI] occupies a neutral space and can transfer information between modalities.
>
>
> ### **Response to Q1 & Q3**
>
> Thank you!
>
> ### **Response to Q2**
>
> We will make explicit that A_{img->text} refers to a submatrix and not to a scalar using the slicing notation, which will make it more transparent which columns are selected. This is an update of line 88:
>
> *We define $A_{\text{img} \to \text{text}} = A_{\left[N_{\text{EOI}} + 1: N,  0: N_{\text{EOI}} -1 \right]}$, where the notation $A_{[a:b,c:d]}$ denotes the submatrix consisting of rows from a through b, and columns c through d inclusive.*
>
> [1] Ge et al., *Planting a SEED of Vision in Large Language Model*;
>
> [2] Wu et al., *Towards Semantic Equivalence of Tokenization in Multimodal LLM*.

---

> > ### Comment · Reviewer_7J47 · 2025-08-05
> >
> > I appreciate the author's thorough responses. Most of my major concerns are resolved, and I believe the author's additional experiment regarding my W1 provides significant insights for future works in MI. So I chose to raise my score to 5.

---

### Comment · Area_Chair_89jf · 2025-08-02
**Discussion Phase**

Dear Authors and Reviewers,

I would like to thank the authors for providing detailed rebuttal messages. I would also like to thank reviewer  7J47 for already engaging in further discussion!

For the other **reviewers**, I would like to encourage you to carefully read all other reviews and the author responses and engage in an open exchange with the authors. Please post your first response as soon as possible within the discussion time window, so there is time for back and forth discussion with the authors. All reviewers should respond to the authors, so that the authors know their rebuttal has been read.

Best regards, AC

---

### Note · Authors · 2025-08-14

We thank all reviewers for the stimulating discussion and the Area Chair for coordinating a productive review process.\
Below, we summarize the strengths of our work highlighted during the discussion and explain how we addressed the reviewers’ concerns.

**Strengths acknowledged before the rebuttal.**

Before our rebuttal, reviewers agreed that our main finding, namely that *multimodal-output VLMs rely on a single token for image-to-text information transfer* (mkQ5), was novel (7J47), surprising (9UB8), and well-supported by an extensive (R1ay, mkQ5, gNmN) and carefully designed set of experiments (9UB8, gNmN). \
They found our explanations clear (9UB8, R1ay, gNmN) and the analysis a reliable *groundwork for future interpretability studies of unified VLMs* (7J47, gNmN) with broad implications for model *training* (9UB8), *control, interpretability, and robustness* (mkQ5).

**Our responses to reviewers’ concerns.**

**Practical applications.**
Several reviewers (7J47, 9UB8, R1ay, gNmN) asked about the practical relevance of our findings. \
We addressed this concern with an experiment suggested by reviewer R1ay based on a claim in our original submission (l. 326-327). Reviewers found this experiment *very interesting*, providing *additional evidence for our interpretation of the results* (gNmN) and the *potential role of the narrow gate in VLMs* (7J47). It also showed *actionable outcomes for practitioners* (9UB8).

**Robustness of our methods.** Other questions challenged the robustness of our techniques for analyzing information flow in VLMs: the validity of our zero-ablation approach (7J47), and the generality of our attention analysis (9UB8) and semantic probe (9UB8, R1ay).\
We showed that our findings remain robust when changing the ablation strategy (7J47) or focusing the attention map analysis on answer tokens (9UB8). Importantly, we showed that the semantic analysis initially based on the ImageNet classification task also *generalizes to other tasks* (9UB8), and we *thoroughly verified [our results] across multiple tasks, benchmarks, and baselines* (R1ay).

**Interpretation and novelty.** We gave an interpretation of the mechanism underlying the formation of the narrow gate (7J47), and clarified why our contribution extends beyond prior literature (mkQ5).

**Outcome**

Reviewers who responded fully to our rebuttal considered their concerns addressed and updated their evaluations accordingly.

---

### Decision · Program_Chairs · 2025-09-17

**Decision:**

Accept (poster)

**Comment:**

The paper shows that multimodal-output VLMs concentrate image/text communication through a single token (narrow gate), while unimodal-output VLMs communicate in a distributed way. Ablating the EOI token cripples performance and editing it changes semantics. The rebuttal adds broader evidence and a small SFT-masking study.

The clear strength of this paper is that it presents is an increased understanding of how information flows in the investigated models. Reviews praised a careful analysis that links attention to causal effects with multiple probes. After the discussion, there was also a broadened validation across tasks/datasets, added training intervention with promising outcomes.

One reviewer remains borderline-reject desiring improved actionability and narrative. The exact mechanism behind the gate formation could also benefit from further investigation. There were some issues regarding clarity or formatting issues.

In the discussion two reviewers increased their scores while one reviewer stayed slightly negative but not arguing strongly against acceptance.

Overall, I recommend to **accept** this paper to NeurIPS. It analyses an interesting topic in a way that would be very suitable for presentation in my opinion.